# Laboratory evidence of magnetic reconnection hampered in obliquely interacting flux tubes

Simon Bolaños[1,2], Andrey Sladkov[3], Roch Smets[2], Sophia N. Chen[4], Alain Grisollet[5], Evgeny Filippov [3,7], Jose-Luis Henares[8], Viorel Nastasa [4,9], Sergey Pikuz [6,7], Raphël Riquier[5], Maria Safronova[3], Alexandre Severin[1], Mikhail Starodubtsev [3] & Julien Fuchs [1] ✉

Magnetic reconnection can occur when two plasmas, having anti-parallel components of the magnetic field, encounter each other. In the reconnection plane, the anti-parallel component of the field is annihilated and its energy released in the plasma. Here, we investigate through laboratory experiments the reconnection between two flux tubes that are not strictly anti-parallel. Compression of the anti-parallel component of the magnetic field is observed, as well as a decrease of the reconnection efficiency. Concomitantly, we observe delayed plasma heating and enhanced particle acceleration. Three-dimensional hybrid simulations support these observations and highlight the plasma heating inhibition and reconnection efficiency reduction for these obliquely oriented flux tubes.

Magnetic reconnection is the subject of intense investigations due to its suspected role in the sudden plasma heating and particle energization observed in varied spatial and astrophysical events, e.g., solar flares[1–3], planetary magnetic substorms[4–6], flux transfert events[7,8], or black hole plasma jets[9]. However, this phenomenon remains difficult to characterize and measure with sufficient resolution in all these distant events. As such, many complementary laboratory experiments have been developed to investigate in detail this process, using various experimental platforms[10–13]. However, despite such experimental effort and continuous theoretical developments from many groups[14–19], persistent difficulties remain, including being able to predict accurately the observed fast onset of magnetic reconnection.

A factor complicating the picture is the topology of the magnetic fields. Deviating from the idealized picture of the first theoretical models[20], in which investigations were restricted to the reconnection of anti-parallel magnetic fields (e.g., the components of the incoming magnetic field lie in the $yz$ plane), the magnetic fields in natural events[21,22] can be decomposed into the anti-parallel components and possibly an additional component (e.g., along the $x$-axis) of the magnetic field (commonly called a guide field). A well-known example is that of reconnection in solar arches[23]. It involves the emergence of magnetic flux tubes from the convective zone at the solar surface[24]. These flux tubes can meet and then reconnect. The simplest reconnection picture involves the encounter of two straight flux tubes exhibiting anti-parallel magnetic field. In reality, the situation is rather three-dimensional, either due to the oblique orientation of one flux tube with respect to the other (the configuration explored with our experimental set-up, see below) or due to the twist of the flux tubes along their axis, resulting in the growth of a poloidal component of the magnetic field[25] (associated with a current flowing along the core field). The pending challenge the community faces here is to understand how does reconnection take

[1]LULI - CNRS, CEA, UPMC Univ Paris 06 : Sorbonne Université, Ecole Polytechnique, Institut Polytechnique de Paris, F-91128 Paris, Palaiseau cedex, France. [2]LPP, Sorbonne Université, CNRS, Ecole Polytechnique, F-91128 Palaiseau, France. [3]Institute of Applied Physics, 46 Ulyanov Street, 603950 Nizhny Novgorod, Russia. [4]ELI-NP, Horia Hulubei National Institute of Physics and Nuclear Engineering, Bucharest, Magurele, Romania. [5]CEA, DAM, DIF, F-91297 Arpajon, France. [6]National Research Nuclear University MEPhI, 115409 Moscow, Russia. [7]Joint Institute for High Temperatures, RAS, 125412 Moscow, Russia. [8]Centre d'Etudes Nucléaires de Bordeaux Gradignan, Université de Bordeaux, CNRS-IN2P3, Route du Solarium, F-33175 Gradignan, France. [9]National Institute for Laser, Plasma and Radiation Physics, Magurele, Ilfov, Romania. ✉e-mail: julien.fuchs@polytechnique.edu

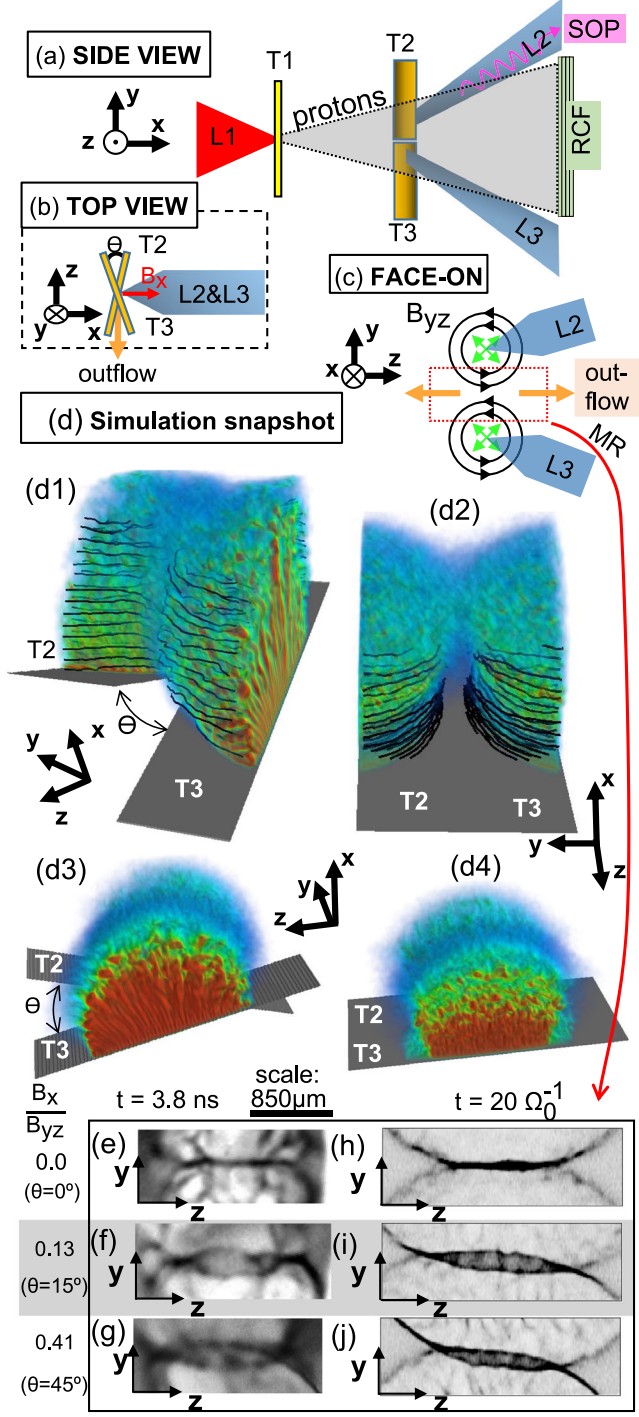

**Fig. 1 | Setup of the experiment and simulations, and proton radiography results.** Three schematic views of the experimental setup, along three projections. The purple wavy arrow in **a** indicates the direction in which the optical self-emission is recorded, while the orange arrow in **b** indicates the direction of the particle spectrometer in which the ejected particles (ions and electrons) are analysed. The red arrow in **b** shows the guide-field, which magnitude is varied by changing the angle $\theta$. **d** Snapshots of a simulation in the tilted configuration (d1-d2) and in the coplanar configuration (d3-d4) using the 3D hybrid code AKA[53]: the targets are depicted in grey, and electron density in color, and the black lines represent the magnetic field lines. **e–j** Zooms in the MR region (illustrated by the red dashed rectangle in **c**) of the experimental proton deflectometry images (with respect to time $t$, $t = 0$ refers to the start of the laser irradiation), for various strengths of the guide field ($B_x/B_{yz}$ gives the ratio of the guide field vs. the in-plane magnetic fields). **e–g** Experimental images (snapshots integrated over ~3ps), toward the end of the laser irradiation (which lasts for 5 ns). Note that the spatial scale at the top applies only to panels **e–g** and is relative to the target plane. **h–j** Corresponding synthetic images of proton-deflectometry obtained in the simulations, at a time comparable to that of the experiment, i.e., once we are in the fast linear growth phase of the reconnection event. In both the experimental and synthetic deflectometry images, the dark regions correspond to an accumulation of protons.

However, conflicting results have been reported when investigating guide-field reconnection. For strong guide field, i.e., when its strength is comparable or larger than the one of the anti-parallel in-plane magnetic fields, opposite results have been highlighted, with numerous simulation studies showing that reconnection is quenched[31–33], but one observation reporting conversely that the guide-field might aid fast reconnection to take place[22]. For weaker guide field, the reconnection rate has not been evoked to be affected[34,35], or only weakly so[33].

Here we show, using laboratory measurements, which we compare to three-dimensional hybrid simulations, how even a weak angle between two flux tubes strongly affects the rate of a reconnection event. As shown in Fig. 1, we use two symmetric and adjacent magnetized plasma plumes, which axis of symmetry are either parallel or tilted. To generate these plasma plumes, the experiment uses two 200-J energy and 5-ns duration lasers beams (L2 and L3 in Fig. 1a), irradiating two 5-$\mu m$ -thick copper foils at an intensity of $2.5 \times 10^{14}$ W/cm$^2$ (see Methods). The lasers create two hot, dense adjacent plasmas (separated by $500 \, \mu m$), expanding toward each other supersonically (at ~81 km/s)[36] for all the laser pulse duration. In each blow-off plasma, a magnetic field is generated by the crossed density and temperature gradients induced by the laser (i.e., the Biermann-battery effect)[37]. The overall topology of each magnetic field is analog to a flux tube connected to itself (see Fig. 1c). Magnetic reconnection ensues from the encounter of the two magnetic structures. The in-between zone is the current sheet associated with the reversal of the magnetic field. Such experimental set-up allows studying the reconnection between two flux tubes, with a tunable relative orientation.

## Results

Following the methodology outlined in Ref. 36, we model using the FCI2 code[38] the magnetic field generation, matching the measurements performed using proton radiography (which are detailed below). Close to the center of the plasma, the magnetic field is compressed toward the target by the Nernst effect[36,39]. However, toward the edge, i.e., where the two plumes will interact, the magnetic field is advected with the plasma flow and expands also away from the target (along the $x$-axis)[40], resulting in a height of few tens of ion inertial lengths. The magnetic field strength in each of the plasma plume is ~300 T. The three-dimensional simulations detailed below mock up this magnetic structure, as illustrated in Fig. 1d.

When the two plasmas are driven from co-planar targets, i.e., $\theta$ (as defined in Fig. 1b) is equal to 0, the magnetic fields embedded in each plasma plume are anti-parallel, leading to anti-parallel merging. As a result of the convergence of the two plasmas (in less than 1 ns),

place between two obliquely oriented flux tubes, which includes the presence of a guide field.

Likewise, when leaving the solar corona and cruising toward the Earth, magnetic flux ropes (that is twisted flux tubes) can interact with the Earth magnetopause and also eventually reconnect during Flux Transfer Events (FTE[26]). As the signature of the bi-polar component of the magnetic field, normal to the magnetopause, can be clearly identified, magnetic reconnection between two obliquely oriented flux tubes has been put in evidence both by in-situ observations[8,27,28] and numerical simulations[29,30]. The modified efficiency of the reconnection process between two obliquely oriented flux tubes then also appears to be a key point in explaining how mass, momentum and energy can be loaded in the inner Earth magnetosphere (or in other planets[5] like Mars) from the solar wind flowing around.

magnetic reconnection takes place[12,41–45]. Now, by tilting one target with respect to the other by the angle $\theta$, an out-of-plane guide magnetic field ($B_x = B_{yz} \tan \theta / 2$, illustrated by the red arrow in Fig. 1b) naturally arises in the plasmas encounter region. The strength of this guide field ($B_x$) is controlled by modifying $\theta$ (see Fig. 1b). Note that the effective distance between the two plasma plumes will increase when tilting the two targets. However, in our conditions, this distance is only increased by less than 7% at x = 230 $\mu$m (or x = 25$d_0$ in the simulations) for 45° tilt compared to the separation between the two laser impacts in the coplanar configuration. Such modest increase is much lower than any quantitative change observed in the experimental and simulated observables. We further stress that the tilt does not significantly affect the interaction volume between the two flux tubes, as will be detailed below.

Note that as a result from this tilt angle, the strength of the in-plane magnetic field ($B_{yz}$) is reduced at most of 8%, which is less than the shot-to-shot fluctuations. Note also that the guide field is here inhomogeneous, which is different than the setup of most numerical studies[31,33,35,46]. There is however no clear reason why the out-of-plane magnetic field would be uniform in some natural events and not more alike our experimental setup (see e.g. in ref. 3). Although uniform guide-field is relevant to some reconnection event[47], previous numerical studies used a uniform guide field as a numerical simplification.

We will now detail the features observed in the experiment and identify the processes in place. Our method for diagnosing the time-resolved spatial distribution of the magnetic fields uses fast, laminar protons[48] induced by a high-intensity laser beam (L1, illustrated in Fig. 1a). These protons are deflected by the magnetic fields as they propagate through the target assembly shown in Fig. 1, after which they are collected on films (RCF, see Methods). Hence, analyzing the deflectometry patterns yields information on the fields encountered along their way by the probing protons. The films shown in Fig. 1e–g display the dose modulations impaired on 14 MeV protons for various experimental configurations, for coplanar and tilted targets. The main information regarding the interaction between the two magnetic loops is contained in the proton deflectometry pattern at the midway between the two plasmas (namely the current sheet), since this pattern is the result of the deflections induced by the path-integrated magnetic field. We can observe in the experimental, as well as synthetic (computed from the simulations), proton-deflectometry images shown in Fig. 1e–j two main patterns: (A) a black thin line (see Fig. 1e and h), and (B) a "mouth" shape pattern, seen in Fig. 1f, g and i, j.

The probing protons traversing the magnetic fields zone are deflected outward (see Methods). The Lorentz force deflection imposed on them thus induces a lower dose of protons in the zones of strong magnetic fields, and a concentration of the protons at the edge of the magnetic field zone. This is what is observed in the films, both exhibiting experimental and synthetic whiter zones (of less proton dose) within, and an outer darker rim. Pattern (A), the black thin line in the proton image in the diffusion region between the two magnetic domains (see Fig. 1e and h) shows that the probing protons undergo only local deflections, i.e., the deflections are smaller that the length scale of the diffusion region. Conversely, pattern (B) is produced by an "overshoot" of the proton deflections: the protons are no more just pushed at the edge of the zone, but are subject to a further deflection that pushes them beyond the boundary of the opposite domain. This, therefore, attests of an increased gradient of the magnetic field at the reconnection site.

Since patterns (A) and (B) can be respectively identified in the coplanar and tilted targets case, we can conclude the following: in the coplanar case, reconnection takes place and we do not witness a pile-up of magnetic field, while in the tilted case, where a guide field arises,

an increased gradient of magnetic field is observed in the reversal region. The former is consistent with what was previously observed[49] in a similar coplanar geometry and with similar laser parameters. Conversely, in the tilted case, since we observe an increased magnetic field gradient, we can conclude that reconnection is less efficient. This stems from the fact that, in the coplanar and tilted cases, the same amount of magnetic field is created and converges toward the current sheet. However, since we observe in the tilted cases that the magnetic field doesn't evacuate as efficiently as in the coplanar case, it, therefore, means that it accumulates at the current sheet.

We stress that the proton radiographs shown in Fig. 1 are only a sub-sample of the data collected during the experiment. The analysis detailed above is based on a larger set of data, which supports the reproducibility of the features discussed here.

The self-emission in the optical domain (see Methods) observed to originate from the hot plasma in the reconnection region (represented by the orange box in Fig. 1c), and shown in Fig. 2, concurs with the observation of a less efficient reconnection when the guide field arises. The self-emitted light is here recorded across the current sheet, close to $z = 0$, along the symmetry axis of the two plasmas (the $z$-axis), and streaked in time. The emitting plasma is in an optically thin regime, meaning that the self-emission increases with the plasma density, but decreases with the plasma temperature (see Methods). When only one laser beam (either L2 or L3) irradiates the target assembly, we observe in the reconnection region, i.e., 250 $\mu$m away from the laser spot, a quite steady self-emission over time (see the solid line in Fig. 2). This takes place with a slight delay (~0.3 ns) compared to the start of the laser irradiation, this delay being due to the need for the plasma to expand laterally up from the laser energy deposition spot to the location of observation.

The overall behaviour is quite different when the two laser beams irradiate the targets. At the onset of the self-emission, we observe a first increase of the self-emission compared to that induced by one laser beam. It is followed by a plateau, and, later, a decrease (see Fig. 2d). We interpret the first, fast increase as due to the increased density in the reconnection region induced by the pile-up when the two expanding plasmas compress the current sheet (before the onset of the reconnection). The later decrease of the self-emission in the two beams case is likely due to two cumulated factors: (i) as reconnection takes place, the accumulated plasma can be evacuated from the reconnection layer and hence its density decreases, and (ii) the increased temperature of the plasma as the magnetic energy is transferred to the plasma. Without guide field, this emission decrease is seen in Fig. 2d to take place much more rapidly, i.e., around 1 ns. This is consistent with the onset of reconnection taking place around that time in this case. In the presence of a guide field, we notice three changes: (1) the first increase in the plasma emission is enhanced, (2) the plateau lasts longer and the decrease takes place later, and (3) the delay in the latter increases with the guide-field strength (compare the dashed and dotted-dashed lines in Fig. 2d). All this is well-consistent with the less efficient reconnection of the magnetic field observed in Fig. 1f–h when a guide field is associated.

Figure 3 shows that in a presence of a guide field, strong ion and electron spectra (see Methods) are also recorded being ejected along the current sheet axis (the $z$-axis), i.e., along the expected outflow direction (as indicated in Fig. 1). Quite differently, no signal in both channels can be recorded above the noise level in the coplanar case, or when looking in the perpendicular direction (along the $y$-axis).

The absence of signal in the coplanar case is not so surprising: since our spectrometer looks in the $yz$-plane, along the target surface, it would miss the particles that are accelerated along the $x$-axis. This is likely the case for most of the particles accelerated following

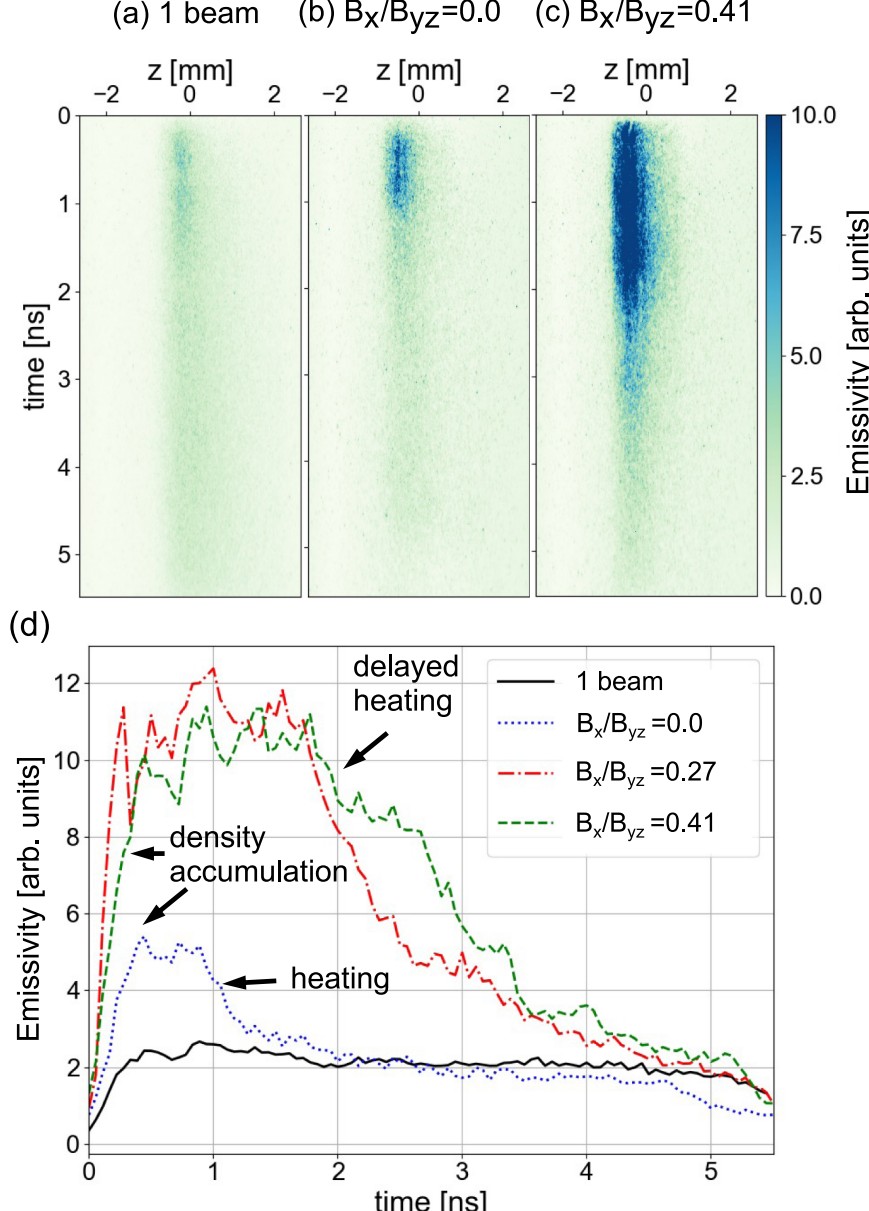

**Fig. 2 | Laboratory optical pyrometry observation of delayed heating in the reconnection area and in the presence of a guide field. a–c** streaked images of the self-emission of the plasma in the reconnection region as recorded in the laboratory experiment along the *z*-axis (see Fig. 1), in the mid-plane between the irradiation spots of the L2 and L3 lasers. The self-emission is recorded for photons around (470 ± 135) nm wavelength and integrated over 230 μm along the *y*-axis (see Fig. 1). Panel **a** corresponds to only one laser beam (L2) turned on. Panels **b** and **c** correspond to the case where L2 and L3 are fired; for panel **b** on coplanar targets, while for panel **c** the targets are tilted, resulting in the presence of a $B_x/B_{yz} = 0.41$ guide field. For all, we observe that at late times, past 5 ns, the signal dies out. Indeed, as the laser is switched off, the heat flux cannot maintain the magnetic field anymore. As a consequence, the dense plasma inducing the observed emission cannot be confined anymore in the magnetic field and quickly expands into vacuum. **d** Lineouts of z-integrated streaked plasma self-emission and as a function of time of the images shown in **a–c**, plus of another shot corresponding to an intermediate guide field strength. Time *t* = 0 corresponds to the start of the targets irradiation by L2 and L3.

reconnection: as the particles in the plasma inflows reach the reconnection area, they will be influenced by the $E_x$ component of the electric field. Hence, this component of the electric field drives the particles in the *x* direction[50,51]. The interesting point is that a strong signal is seen in the presence of a guide field, i.e., when there is significant magnetic field pile-up. These particles are thus likely not accelerated during reconnection, but ahead of the actual reconnection, either through slingshot Fermi acceleration[50] or betatron acceleration[51,52]. Since these mechanisms would obviously become more efficient if there is accumulated magnetic field powering the acceleration, the observation of energization in the tilted case is

applied is well compatible with the observation of piling-up of magnetic field in that configuration compared to the coplanar one.

## Discussion

The appropriate numerical tool for end-to-end full-scale simulations of the experiments on magnetic reconnection has to address the well-known duality between a large-scale problem (to properly treat the inflow and outflow regions without arbitrary boundary conditions) and its microphysics description on at least the ion scale. Therefore, we chose to use the three-dimensional hybrid code AKA[53,54] (see Methods). This code solves the ion kinetic dynamics following the PIC formalism

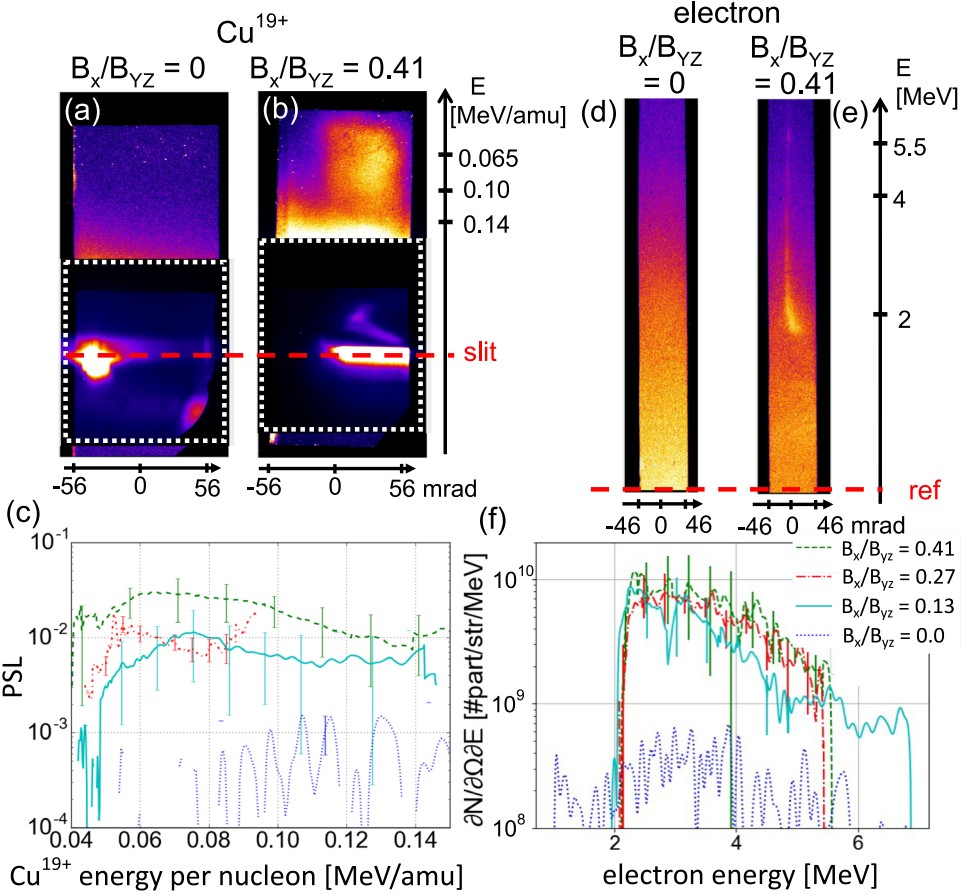

**Fig. 3 | Laboratory evidence for enhanced particle acceleration along the current sheet in the presence of a guide field.** Raw spectra of Cu ions as recorded by the particle spectrometer in the absence (**a**) or presence (**b**) of a guide field. Since the T2 and T3 targets are made of Copper, note that we made the assumption that the ions recorded in the spectra are $Cu^{19+}$, based on the average ionization state we expect to have in our plasma conditions. The vertical axis is the spectral one, the horizontal one is sampling the angle of emission of the particles from the targets. For panel **b**, we have $B_x/B_{yz} = 0.41$. In the region surrounding the projection of the spectrometer entrance slit onto the detector, the images are strongly saturated by the X-ray and visible light emitted from the plasma. This is why the images shown are a patch of the first scan of the image plate detector (top part) where the ion signal can be seen, with the second, unsaturated and delimited by the white dashed line, scan of the detector. Because of the noise close to the slit, the spectrometer was reliably able to detect ions only in the shown energy range of 0.04 keV/amu to 0.16 keV/amu. **c** Lineouts of the spectra such as shown in **a**, without guide field (dotted line) or with a guide field of various magnitude (full, dashed, and dotted-dashed lines, see panel f for the legend). Each plot results from the integration over the angular dimension and is obtained by averaging two to five shots, depending on the configurations, recorded in the same conditions. The error bars correspond to the standard deviation of the signal over these shots. (**d**–**f**) Same as **a**–**c** for the electrons. We note that the recorded energies are much higher than if the particles would be ejected merely at the Alfvén velocity. However, this is not surprising as already many numerical and experimental studies have shown that higher energy gain could be expected[80–83].

and describes electrons by a 10-moments fluid ($n$, $\mathbf{V}_e$, $\mathbf{P}_e$). We consider in the simulation the same coplanar and tilted cases used in the experiment. To mimic the ablation process, we include a heat operator that pumps electron pressure in the surface region, and a particle creation operator that sustains the constant solid target density. These two operators allow an axial electron density gradient and a radial electron temperature gradient to be created and sustained. As a result, a Biermann-battery magnetic field is continuously produced[36,39]. Figure 4a, b illustrate this $B_z/B_0$ field in the $xy$-plane i.e., orthogonal to the laser axis and passing at $z = 0$ between the two plasmas. To reach the curvature radius of the experimentally observed magnetic fields and save computational resources, we consider an elliptical profile for these two operators. The major semi-axis of the resulting structure is $34d_0$ (where $d_0$ is the ion inertial length) while the minor is $23d_0$, the resulting curvature radius in the reconnecting current sheet being of the order of $50d_0$.

What we observe in the diffusion region when increasing the tilt between the targets is the growth of an out-of-plane component of the magnetic field. Concomitantly, as will be detailed below, we also observe: (1) an increased magnetic field gradient in the magnetic field

reversal region (see Fig. 4), which is linked to (2) a reduced current sheet thickness (see below), (3) an increased electron density in the same region (not shown here), and (4) a decreased electron temperature in the reconnection region (see Fig. 5b). We also point to a (5) reduction of the magnetic reconnection efficiency (see Fig. 5a), in consistence[55] with an observed (6) decreased plasma $\beta$ parameter (i.e., the ratio of electron pressure to magnetic field pressure) at the field reversal (see Fig. 6a–c). We now detail the observations in the simulations that concur with the experimental ones.

We first focus on the increased magnetic field gradient observed when tilting the axis of the plasma plumes (see Fig. 1). The stronger magnetic field gradient around the field reversal for increasing tilt, especially close to the targets ($x = 0$), can be seen by comparing Fig. 4a and b (see the narrower $B_z = 0$ region around $y = 0$ in the tilted case). These correspond, respectively, to the coplanar case ($\theta = 0°$) and the tilted case ($\theta = 15°$). The field gradient is further seen in the profiles of $\int dx(B_z)$ that are plotted in Fig. 4c for the two considered cases (see the sharper gradient of the magnetic field reversal around $y = 0$ for the $\theta = 15°$ case). The sharper gradient in the tilted case then drives the mouth-shaped structure of the proton radiography, both observed in

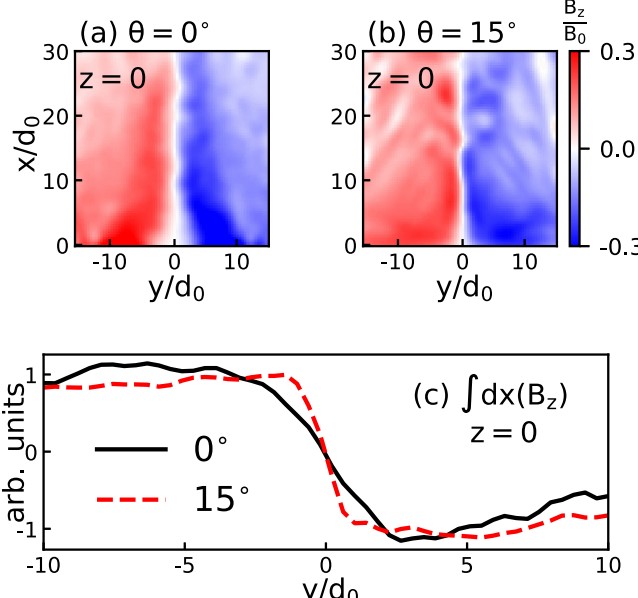

**Fig. 4 | Simulated evidence for magnetic field compression in the presence of a guide field.** Central cut of the anti-parallel component of the magnetic field (in the $xy$ plane, see Fig. 1 for the geometry), as retrieved from the 3D simulations, for $\theta = 0°$ -(**a**) and $\theta = 15°$ -(**b**) at $t\Omega_0 = 20$, where $\Omega_0$ is the ion gyrofrequency (see Methods). This represents the $B_z$ field across the contact region between the two counter-streaming magnetized plasma plumes. Panel **c** displays the path-integrated (along $x$) $B_z$-field passing by the contact region at $t\Omega_0 = 20$.

the experiment and in the simulations, see Fig. 1. We observe that the sharpening of the magnetic gradient is induced by a reduction of the current sheet thickness (point (4)) that can be seen in Fig. 6d–f.

As in the experiment (see Fig. 2), we also observe in the simulations, when tilting the magnetized plasma plumes, a reduction of the electron heating (see Fig. 5). As follows from our recent study[56] of symmetric low-$\beta$ reconnection, a decrease of the gradient of the electron inflow velocity (induced by the compressed guide field, generating a current along $y$) is responsible in turn for a less efficient increase of the electron pressure. Thus, in the tilted case, the cumulative effect of the density increase and of the electron pressure reduction results in the decrease of the electron temperature compared to the coplanar case. This is what is clearly observed in the simulation, as shown in Fig. 5b. The scalar value plotted here is straightforwardly the third of the trace of the electron pressure tensor, divided by the electron density. This reduction of the temperature in the tilted cases is in good consistency with the results from the experimental pyrometry diagnostic (see Fig. 2).

We now discuss the observed lower reconnection efficiency in the tilted target case. Aside from a direct evaluation of the reconnection electric field[57], an alternative approach is to focus on the dissipation term $\mathbf{J} \cdot \mathbf{E}$ as usually done with satellites in-situ measurements[6,58] or in laboratory experiments[59]. We here compute this term in the electron frame, which is integrated inside the current sheet, for $|y/d_0|<5$. In order to retrieve only the energy that is transferred from the fields to the particles, we keep the $\mathbf{J} \cdot \mathbf{E} > 0$ component which is displayed in Fig. 5a, in arbitrary units. One clearly observes that the larger the tilt angle (hence the guide-field), the smaller the dissipation. While not a direct proof, this observation strongly supports the reduced efficiency of the reconnection process in the tilted case, which we inferred from the experimental proton radiographs shown in Fig. 1. It is also consistent with previous studies pointing to a slowdown of the reconnection process because of the presence of a guide field[32,33].

We checked that the observed reduced reconnection efficiency is not merely due to a geometrical effect, i.e., we checked that the

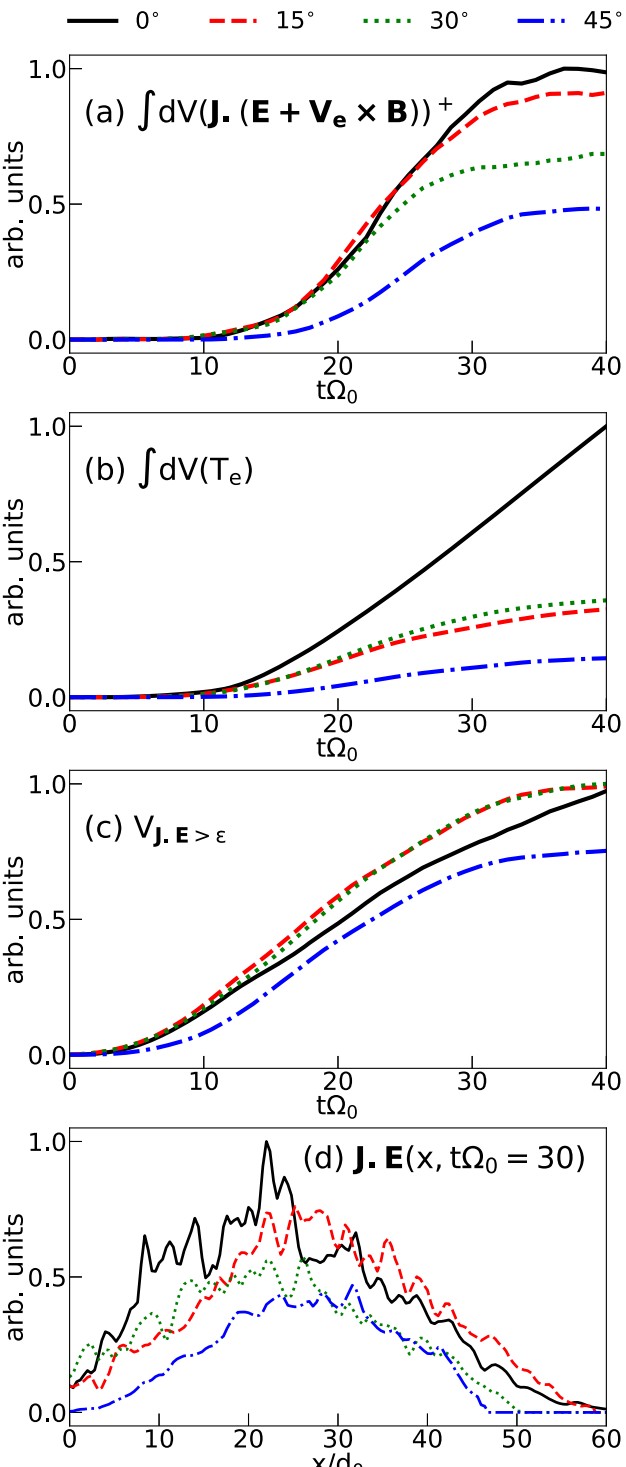

**Fig. 5 | Temporal evolution of various simulated quantities. a** Simulated power of the energy conversion from magnetic energy into plasma energy in the electron frame, which is related to the efficiency of the reconnection process (see text). **b** Evolution of the space-integrated electron temperature retrieved in the simulations, here defined as one-third of the electron pressure tensor trace divided by the electron density. **c** Evolution of the diffusion volume. The diffusion volume is the volume where the integrand from **a** being greater than the numerical noise level, $\epsilon = 10^{-8}$. The integrated volume for the three computed quantities is limited in the y direction over $-5 < y/d_0 < 5$. **d** x-dependency of the reconnection efficiency: dissipation term ($\mathbf{J} \cdot (\mathbf{E} + \mathbf{v}_e \times \mathbf{B})$) integrated over the yz-plane at $t = 30\Omega_0^{-1}$.

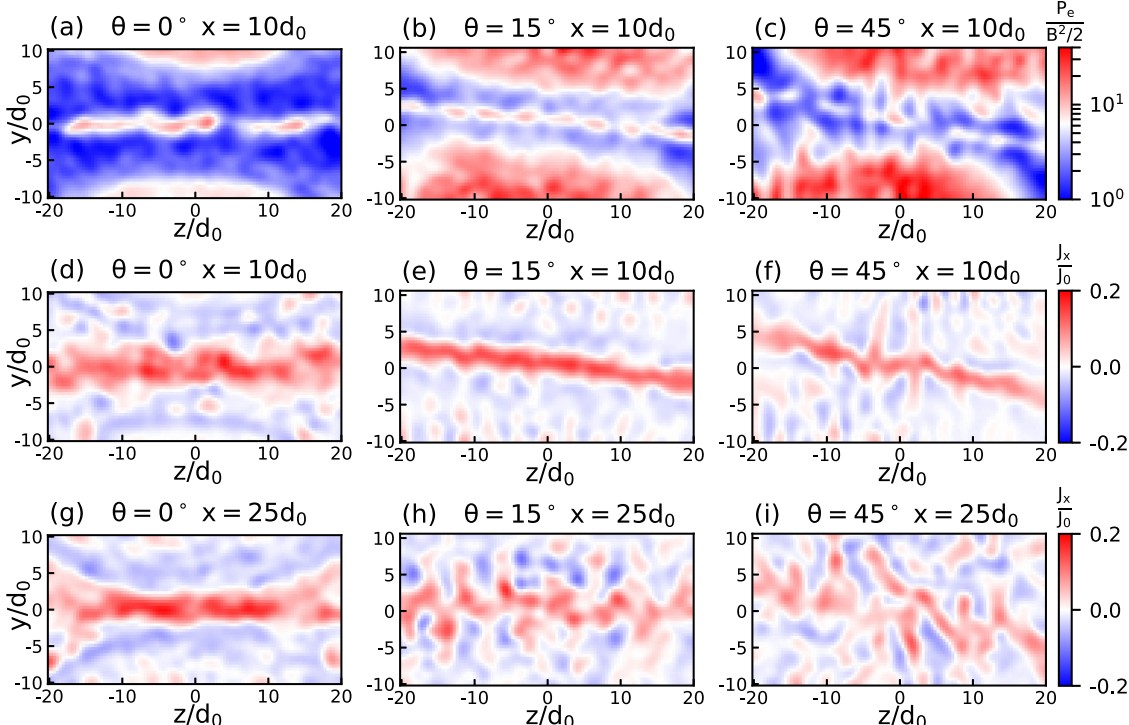

**Fig. 6 | Simulated morphology of the current sheet for various tilt angles and at various heights.** 2D cuts of the electron plasma $\beta$ parameter and out-of-plane current. **a–c** 2D cuts (in the $yz$ plane, $x = 10d_0$) of the electron plasma $\beta$ parameter (the ratio of electron pressure to magnetic field pressure). **d–i** 2D cuts (still in the $yz$ plane) of the out-of-plane current $J_x$ at $x = 10d_0$ for panels **d–f** and at $x = 25d_0$ for panels **g–i**. All the images are retrieved from the 3D simulations, at $t\Omega_0 = 30$.

interaction volume between the two plasma plumes is not sensibly modified as the two targets are tilted. In order to verify this, we computed the temporal evolution of the diffusion volume, see Fig. 5c. It shows that the interaction volume is weakly affected when tilting the targets. Similarly, Fig. 6 shows that the current sheet length is not depending on the $\theta$ angle, except close to the target, where the surface of the target will constrain the length of the current sheet (not shown here). This geometrical aspect is effective for $x < L_{CS}/(2tan(90°-\theta/2))$, where $L_{CS}$ is the current sheet length, which is negligible for the small angle tilt configurations and has an effect for $x < 9d_0$ in the 45° configuration, see Fig. 5d. Figure 5d shows the x-dependency of the reconnection efficiency. In a similar manner to Fig. 5a, we integrated the dissipation term over the yz-plane (as opposed to Fig. 5a, where it is volume-integrated) to assess how the reconnection is dependent on x. The efficiency for the coplanar and small angle tilt (15° and 30°) cases have a similar trend, i.e., it rises as a function of x more or less in the same manner. Note that this is true although the overall efficiency at 30° is reduced with respect to the coplanar case. For the 45°-tilt configuration, we notice that not only is the overall efficiency even more reduced, but it clearly becomes efficient only at larger x values than for the other cases. This loss of efficiency is significant close to the target ($x < 9d_0$); we attribute it to the geometrical aspect discussed above, i.e., the fact that the target tilt reduces, close to the surface of the target, the length of interaction of both plasmas. The numerical observations are further reinforced by the experimental observations that (i) the length of the magnetic interaction zone between the two flux tubes as well as (ii) that of the heated plasma (following reconnection) vary both weakly as a function of the tilt angle between the two targets, see Fig. 1e–g and Fig. 2a–c, respectively. All this is due to the fact that, while there is an axial gradient (i.e., along the x-direction) of the plasma parameters, the associated $B_z$ magnetic field is weakly depending on the x location. It is so because the height of the plasma plume, at its edge[40], is larger than its radius, so each flux tube is facing its twins with locally comparable values of the plasma parameters. As a

consequence, and as evidenced in both the experiment and the simulations, even in the tilted cases, the length (along the $z$ axis) of the current sheet is weakly depending on the x coordinate, whatever the $\theta$ angle. Although we do not observe any difference in the length of the current sheet for the various configurations, however, as one moves farther away from the target surface, the current sheet becomes more unstable in the tilted configuration, see Fig. 6g–i. This is caused by the fact that close to the top of the plasma plumes, the plasma advection is mainly oriented normal to the target. This leads to less magnetic flux being advected toward the current sheet, thus reducing the magnetic compression and magnetic pressure, as seen in Fig. 4. With a reduced magnetic pressure, the current sheet is more prone to other processes such as shear-flow instability[60].

In addition to the dissipation term discussed above, previous work[55] proposed, in order to gauge the efficiency of the reconnection, to scale the reconnection rate for a Biermann-mediated reconnection proportionally to the electron $\beta$ plasma parameter. Figure 6a–c shows the computed electron $\beta$ plasma parameter for the coplanar and titled configurations. The $\beta$ parameter is ~10 at the field reversal for the coplanar case and decreases twice for the $\theta = 45°$ simulation. While the display in Fig. 6 is limited to $-20 < z/d_0 < +20$, we computed the z-extension of the current sheet as well as its average y-thickness, in order to compute their ratio, which is known as the aspect ratio of the current sheets. It clearly appears that this aspect ratio is smaller than 50, meaning that this current sheet is not unstable to secondary island[61], whatever the tilt angle. Figure 1i–j exhibit not only quite comparable current sheet length in all cases, but also close values of the associated current density $J_x$ because, as discussed above, the proton-radiography is a direct measure of the magnetic field compression.

In summary, by means of a laboratory experiment and three-dimensional hybrid simulations, we have put forward how, in a very reproducible manner, magnetic reconnection becomes less efficient when increasing the tilt angle between two flux-tubes. The

experimental observations point, when increasing the angle between the two targets, to an enhancement of the magnetic compression of the current sheet, and an associated larger electron density and smaller electron temperature in the current sheet. 3D hybrid-PIC simulations concur with the observations and highlight that these features come together with a drop of efficiency of the reconnection process when increasing the angle between the targets, i.e., the angle between the two flux tubes.

We note that the tilted configuration we explored here is relevant to Flux Transfer Events at Earth's magnetopause[7], and to reconnecting solar arches[3,62]. Our findings suggest that the effects of even a weak angle between the two flux tubes is very significant in reducing the efficiency of reconnection in such natural environment. However, as of yet, in these natural events, the influence of such a tilt angle is difficult to assert solely from observations, and thus no conclusive answers could be as of yet drawn on the mechanisms underlying the observations. In this frame, our results could allow to improve the understanding of these reconnection events by adding constrains to their analysis.

## Methods

### Laser experiment

The experiment was performed using the LULI2000 laser facility at the LULI laboratory (France). Three laser beams are used: beams L2 and L3 (see Fig. 1) are 5-ns long (square temporal shape). These two beams irradiate simultaneously (within 100 ps) two Copper foils of 5-$\mu$m thickness. As shown in Fig. 1, the two foils could be tilted, one by $\theta/2$, the other by $-\theta/2$, such that overall the azimuthal magnetic flux tube created on one target was tilted by an angle $\theta$ with respect to the other magnetic flux tube created on the other target. Such configuration allows us to have, as shown in Figure 1, along the x-axis, a component of the magnetic field that is out of the reconnection plane (the yz plane), leading to the generation of a guide field. We use this setup to generate the guide field rather than an externally imposed magnetic field[63] as the latter would limit us to very weak (tens of T) fields. We studied the influence of the guide field by comparing four different cases: co-planar ($\theta = 0°$), 15°, 30°, and 45° which correspond respectively to the following ratios $B_x/B_{yz}$ of the guide field strength ($B_x$) over the one of the anti-parallel magnetic fields ($B_{yz}$): 0, 0.13, 0.27, and 0.41. The two laser focal spot foci are separated by 500 $\mu$m for the data shown in Fig. 1. Each laser beam has a 1.064 $\mu$m wavelength and 200 J of energy. They are equipped with random phase plates[64] in front of the focusing lenses, such that the focal spot is 80 $\mu$m in diameter with an uniform intensity distribution. As a result, the intensity on the solid targets was $I$~$2.5 \times 10^{14}$ W/cm². The initial condition of the experiment, i.e., each spot of laser-target interaction, is modelled using the hydro-radiative FCI2 code[38]. The conditions of the simulations are experimentally anchored using time-integrated x-ray emission produced by the plasmas and recorded during the experiment. This emission is registered by a pair of focusing spectrometers (FSSR)[65]. Each spectrometer used a spherically bent mica crystal with parameters $2d = 19.9376$ Å and radius of curvature R = 150 mm. They were implemented to measure the x-ray spectra of multi-charged Copper ions in the range of 9.0–9.5 Å (1300–1380 eV), in the second order of reflection, with a spatial resolution of about 35 $\mu$m. The x-ray spectra were recorded on passive detectors, namely TR Fujifilm Image Plate[66], protected from optical radiation by thin Polypropylene filters (1 $\mu$m thick) covered a coating of 200 nm thick Al.

### Diagnostic

The main diagnostic is proton deflectometry[67]. It has been implemented in order to observe the expansion of the two magnetic flux tubes induced by L2 and L3 irradiating targets T2 and T3, and the change in the overall magnetic field topology after the two magnetic flux tubes interact. Since the magnetic fields produced by the L2 and L3 lasers are mostly contained on the surface of the targets T2 and T3[36,39], the probing protons are sent quasi-parallel to the normal of the targets (see Fig. 1a, b), such that the deflections induced by the Lorentz force associated with the probed magnetic fields can be recorded[12,67]. The probing proton dose profile, initially quite uniform[48] after they have left the source target, is modulated due to these deflections, and its projection is recorded on films. As the magnetic fields are oriented anti-clockwise with respect to the normal of the target facing the irradiating laser pulse (L2 or L3)[12,36,39], the probing protons are sent through the back of targets T2 and T3 such that the Lorentz force deflection imparted on them is outward. This leads to the observed evacuation of protons (white area) from the centre of each laser irradiation spot, and to a dark rim at the edge, synonym of proton accumulation there, as can be seen in Fig. 1e–g.

To produce the probing proton beamlet, we took advantage of the picosecond-duration laser beam of the LULI2000 facility (L1 in Fig. 1). The facility provides such laser beam with an energy on target close to 60 J, having a diameter spot of 10 $\mu$m. Hence, its intensity on target is $I$~$1 \times 10^{19}$ W/cm², allowing us to generate a proton source stemming from the rear surface of the foil as accelerated by the Target Normal Sheath Acceleration (TNSA) mechanism[68]. Such a mechanism generates a proton beamlet which energy spectrum is characterised by a 100% dispersion and a maximum energy cut-off of the order of 20–30 MeV, as varying from shot-to-shot. Due to the small source size of such TNSA-accelerated proton beam (of the order of a few microns[69]), the resolution is not limited by the proton source size, but by the multiple Coulomb scattering they are subject to when crossing the solid targets (see below). In practice, the spatial resolution, in the plane of targets T2 and T3, is 120 $\mu$m, which is amply sufficient for our investigation. Another advantage of the TNSA-accelerated proton beam is that the maximum aperture of the proton beam can reach a half-angle of 20°[70], which allows to have a compact system: the source of protons can be close to the object to be probed while ensuring to cover a wide field of vision. For example, the magnetic flux tubes generated by the Biermann-battery mechanism (shown in Fig. 1) are of the order of 1 to 2 mm in diameter. To radiograph them, it was enough to place the proton source at r = 1 cm from the magnetized plasma and the detector was positioned on the other side at l = 9 cm. The magnification of the projection onto the detector, which is used to relate the scale observed onto the detector to the scale in the plasma, is thus M = (l + r)/r = 10.

Due to the short duration of the proton acceleration process (of the order of a picosecond[71]), the temporal resolution is mostly limited by the range of proton energies that are recorded in each of the detector films, that range varying across the film stack. In practice, the films shown in Fig. 1 correspond to 14 ± 0.5 MeV, yielding a temporal resolution of 6.9 ps, which is negligible compared to the nanosecond time-scale of the fields evolution[67]. After propagation through the solid targets, the protons are recorded on a stack of radiochromic films[72] placed 9 cm away from the solid targets. The stack is composed of HD-v2 and EBT3 films[72], with a 12 $\mu$m-thick Al foil upfront to protect the films from light and debris. Note that the energy density of a proton beamlet is negligible with respect of the energy density in the plasma plume driven by the nanosecond beams (L2 and L3 in the cartoons of Fig. 1). Hence, the plasma plumes are not affected by the presence of the propagating protons.

To diagnose the self-emission of the plasma with temporal resolution (as shown in Fig. 2), we implemented an optical system to image the plasma, along the axis z, on the entrance slit of a streak camera (model S20 from Hamamatsu). As pictured in Fig. 1, the imaging of the plasma is performed through the lens focusing the laser beam L2 on target and in reflection of a thin pellicle that is positioned upstream of that lens. A colour glass filter (BG38 from Schott) is used to filter out the short wavelengths of the emitted spectrum; the long wavelength part of the spectrum is equally removed from the measurement due to

the strong decrease of sensitivity of the camera in that region. As a result, we record light in a spectral range of $(470 \pm 135)$ nm. The output of the diagnostic is an image which provides, on one axis, spatial resolution along the current sheet as well as, on the other axis, temporal resolution, as shown on the raw data of Fig. 2a–c. Based on hydro-radiative simulations of our experimental configuration (using the FCI2 code), we analyse that the maximum of emissivity originates from an optically thin plasma: in the reconnection zone, where the electron density is of the order of $1 \times 10^{20}$ cm$^{-3}$, the mean free path of a visible photon is $46.1 \mu m$, which is of the same order than the plasma gradient scale length, which is $20–40 \mu m$. As a consequence, Bremsstrahlung is the dominant emission mechanism in the spectral observation range of the diagnostic[73]. Hence, the emissivity of the plasma can be expressed as: $E_{ff} \sim Z \times n_e^2 /[(T_e)1/2 \times g_{ff} \times \nu]$, where $g_{ff}$ is the velocity averaged gaunt factor[74], and $\nu$ is the spectral bandwidth of the diagnostic.

The particle spectrometry is performed simultaneously on ions and electrons by using different detectors recording positively and negatively charged particles dispersed in a permanent magnet, following a design originally made for experiments performed on the Nova-PW laser[75]. The magnet spectrometer is set to analyze particles leaving the plasma along the current sheet (the axis $z$ in our setup), i.e., in the expected direction of the outflow. The diagnostic relies on the deflection of the particles in a well-known magnetic field. The detectors used in this experiment are imaging plates[76].

## Numerical simulations

In our numerical model, electrons are described in a fluid way, by the ten-moments model. Those ten moments are density $-n$ (equal to the total ion density by quasi-neutrality), bulk velocity $-\mathbf{V}_e$ and the six-components pressure tensor $-\mathbf{P}_e$. To properly describe the decoupling of ions from the magnetic field in the ion diffusion region, we keep the ion description at the particle level. Hence, we use a hybrid code, which is here the code AKA[53,54], built on general and well-assessed principles of previous codes[77] like Heckle[78], with advanced features like an ablation operator and a six-components electron pressure tensor.

The electromagnetic fields are treated in the low-frequency (Darwin) approximation, as we consider that the phase velocity of electromagnetic fluctuations is small compared to the speed of light. Neglecting the displacement current, we then write the electron Ohm's law as follows:

$$\mathbf{E} = -\mathbf{V}_i \times \mathbf{B} + \frac{1}{en}(\mathbf{J} \times \mathbf{B} - \nabla.\mathbf{P}_e) + \eta\mathbf{J} \qquad (1)$$

In Eq. (1), $\mathbf{V}_i$ is the ion bulk velocity, $n$ is the electron density, $\mathbf{J}$ is the total current density equal to the curl of $\mathbf{B}$, and $\eta$ is the fixed-value resistivity. We use the explicit integration scheme for the six-components pressure tensor evolution equation[56]. The magnetic field and the density are normalized to $B_0$ and $n_0$ respectively, the lengths are normalized to the ion inertial length $d_0$ (calculated using the density $n_0$), the times are normalized to the inverse of ion gyrofrequency $\Omega_0^{-1}$ (calculated using the magnetic field $B_0$) and the velocities are normalized to the Alfvén velocity $V_0$ (calculated using $B_0$ and $n_0$). Mass and charge are normalized to the ion ones. The normalization of the other quantities follows from these ones. For the resistivity term, we set $\eta = 0.1$: while it provides a dissipation process at sub-ion scales, it is still smaller than the contribution of the electron pressure tensor, whatever the scale, to the reconnection process.

We use a $280 \times 175 \times 175$ grid corresponding to a mesh size equal to $0.4d_0$ in all directions, the time-step is $5 \times 10^{-3} \Omega_0^{-1}$, and free boundary conditions are set in all directions: damping layer for the evolution equations (Faraday's law and pressure tensor evolution equation) and outflow boundary for particles. For the cyclotron term integration in the pressure tensor evolution equation, we use a

reduced ion-to-electron mass ratio $\mu = 10$ while keeping the relaxation parameter $\tau$[56] on electron timescales ($\tau = 0.01$) for numerical stability.

Continuous plasma production by the laser-target interaction is imitated by the ablation operator which includes the heat operator and the particle creation operator. The ablation operator works in the localized area in the near target surface region corresponding to the laser focal spot. We use an elliptic shape for the focal spots with sizes $34d_0$ for major axis and $23d_0$ for the minor one in order to save computational resources. The chosen parameters allow to approach the experimental curvature radius of the magnetic flux tubes, which is of the order of 50 ion inertia lengths. The distance between the focal spot centers is $56d_0$. A heat operator provides pressure increase in the focal spot, the spatial profile of which is set using a fifth order polynom[79] which is smoothly decreasing from a maximum at the focal spot center to zero at its edges. Ions, in turn, are accelerated by the arising electron pressure gradient. The pressure is increased progressively over time (by steps which are constant) in order to obtain the desired temperature ($T_{spot}^e = 4T_0$ where $T_0 = B_0^2/n_0$). The plasma expansion velocity achieves a maximum of $1.5 V_0$ when the plasma is freely expanding, and then oscillates around $V_0$ until the end of the simulations. The particle creation operator sustains a constant target density ($n_{targ} = 2n_0$), mimicking the reservoir of the solid-density target. Particles are added at a specified cold target temperature ($1 \times 10^{-4} T_0$). The magnitude of the self-generated magnetic field is of the order of $0.25 B_0$.

## Data availability

The data that support the findings of this study are available from the corresponding author upon reasonable request.

## Code availability

The code used to generate Fig. 4 to Fig. 6 is AKA. The code is detailed in the Methods section.

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

## Acknowledgements

We thank the LULI teams for technical support, and Fabio Reale (INAF) and Alexandra Alexandrova (LPP) for discussions and comments. This work was supported by the European Research Council (ERC) under the European Unions Horizon 2020 research and innovation program (Grant agreement no. 787539). IAP RAS members acknowledge the support of the Ministry of Science and Higher Education of the Russian Federation (Agreement no. 075-15-2021-1361). This work was partly done within the LABEX Plas@Par project and supported by Grant no. 11-IDEX- 0004-02 from ANR (France). JIHT RAS and NRNU MEPhI team acknowledges the support of RFBR foundation in the frame of projects #20-02-00790 and the MEPhI Program Priority 2030 (#075-15-2021-1305). The research leading to these results is supported by Extreme Light Infrastructure Nuclear Physics (ELI-NP) Phase II, a project co-financed by the Romanian Government and European Union through the European Regional Development Fund, and by the project *ELI–RO*–2020–23 funded by IFA (Romania). The simulations were performed on resources provided by the Joint Supercomputer Center of the Russian Academy of Sciences.

## Author contributions

R.S. and J.F. conceived the project. J.F., S.B., S.N.C., E.F., J.L.H., V.N., S.P., R.R., M.S., A.Se. and M.S. performed the experiments. S.B., E.F., and S.P. analysed the data, with the help of J.F. The hybrid simulations were performed by A.Sl., with the help of R.S., and discussions with A.G. and R.R. The paper was mainly written by S.B., A.Sl., S.N.C., R.S. and J.F. All authors commented and revised the manuscript.

## Competing interests

The authors declare no competing interests.
