## [Peer Review File · Nature Communications]

Laboratory evidence of magnetic reconnection hampered in obliquely interacting flux tubesReviewers' comments:

Reviewer #1 (Remarks to the Author):

Manuscript NCOMMS-19-26005-T, "Laboratory evidence of the halting of magnetic reconnection by a weak guide field" by S. Bolanos et al., (J. Fuchs; Corresponding Author) presents the results of a laboratory experiment in which magnetic reconnection occurs at the interface of two expanding plasma cylinders produced by laser beams impinging on two Au foils. By rotating the target foils relative to one another, the orientation of the magnetic fields at the interface of the two expanding plasmas are altered, resulting in the presence of an out-of-plane "guide" magnetic field. Diagnostics measuring reconnection profiles and outflows indicate that the reconnection process is slowed and/or inhibited when the target foils are in the rotated configuration.

The title and abstract of this manuscript emphasize the role of guide field in the inhibition of reconnection. Since "guide-field" vs "anti-parallel" reconnection is a topic of significant interest, this study could potentially contribute to an understanding of the differences. I nevertheless have concerns regarding the generality of the results, as discussed below.

While the schematic description of the experiment in Figs. 1a–1c provides some sense of the reconnection geometry, it is only Supplementary Figure 14 that provides a clear (albeit idealized) representation for the nominal configuration where the rotation angle θ vanishes. I would suggest that this figure be included in the body of the manuscript. My interpretation of the figure is that orange circles are cross sections of the two parallel expanding plasma cylinders. When θ does not vanish, the cylinders would no longer be parallel to one another, thereby modifying the geometry of the interaction region. Specifically, there will be inhomogeneity in the out-of-plane (x) direction absent from the $\theta = 0$ case. The presence of this inhomogeneity occurs only for those cases where the guide field does not vanish thereby complicating the role of the guide field on the inhibition of reconnection. Indeed, this change in geometry may be more important than the change in topology referred to in the Abstract (i.e., from 2D to 3D). Note that the importance of a transition to 3D topology is questionable since even in the absence of an initial guide field, a significant out-of-plane B_x develops spontaneously in the anti-parallel configuration as evidenced by the quadrupolar Hall magnetic field pattern in the simulation (Fig. 5b).

Note that it is typical in simulation studies addressing the role of an initial guide field for the magnitude of the reversing magnetic field to be held constant. In the present study, it appears instead to be the total magnetic field that is held constant. Therefore, at least some of the inhibition of reconnection in the presence of the guide field is presumably due to the reduction in the strength of the in-plane reversing magnetic field by a factor of $\cos(\theta/2)$. This could potentially account for the delay in the timing of the spike in the reconnection rate in the simulations (e.g., Fig. 5a vs Fig. 5c). Since the simulations appear to be two-dimensional, they would not be able to address the out-of-plane inhomogeneity in the experimental setup previously discussed. Such 3D simulation studies might be able to account for additional dependencies on the angle θ . As a final note regarding the simulations, it is known that electron kinetics (such as off-diagonal terms in the pressure tensor) can influence the reconnection rate. Such effects are absent in the hybrid simulations described here.

While I do not question the results of the experiments presented in this manuscript, I cannot accept—without further justification—the conclusion implicit in the title that it is the presence of the weak guide field alone that controls the rate and character of the reconnection. Furthermore, the nature of reconnection studied is an example of driven reconnection where the plasma regions in which the reversing magnetic fields are embedded are dynamically forced together. Many reconnection scenarios involve plasmas that are juxtaposed, but otherwise initially stationary so that the reconnection is initialized by means of an instability such as tearing. Caution is needed when applying the conclusions of a particular study to a more general scenario. In consideration of the wide variety of reconnection regimes, please also note that while the references cited in this manuscript do point to representative examples of prior theoretical, numerical, and experimental studies, they do not necessarily convey the vast number and breadth of such studies—including investigations of guide-field dependence.

In light of the above considerations, I believe that the experiment reported in this study is deserving of publication, but only if the conclusions are couched in a less general context. For example, a title such as “Laboratory evidence of the halting of magnetic reconnection in the interface between non-parallel expanding plasma cylinders with embedded azimuthal magnetic fields” might more accurately describe the current experiment. Of course, doing so might also suggest that a more specialized journal than Nature Communications would be preferable as a publication venue.

Reviewer #2 (Remarks to the Author):

Report on NCOMMS-19-26005-T, “Laboratory evidence of the halting magnetic reconnection by a weak guide field”, by S. Bolanos et al.

Magnetic reconnection is an important fundamental plasma process occurring in many astrophysical and laboratory fusion plasmas. There exist a large number of unanswered major problems such as when and how does magnetic reconnection initiate and at what speed does it proceed under various magnetic geometries. This manuscript describes laboratory experiments and numerical simulations designed to address one particular problem on magnetic geometries: dependence of reconnection process on the guide field strength.

However, I have the following two major issues on the main conclusions of this manuscript which prevent me from recommending it be accepted for publication on Nature Communications:

1. The experimental setup is a truly 3D geometry which does not have an invariant direction to qualify as a 2D process. The assumed reconnection takes place in the reconnection plan (the y - z plan) and is very

localized in the third direction (the x direction). This is in contrast to the simulation setup which uses a truly 2D geometry where the out-of-the plane direction (the x direction) is the invariant direct, i.e., the reconnection process takes place in an exactly same fashion at all x locations from negative infinity to positive infinity. The commonality between experiment and simulation is only about the third component of magnetic field vector, or the guide field component (the x component). Without a guide field, the reconnecting magnetic field vector remains in the reconnection plane (the y-z plan) while with a finite guide field, the magnetic field vector has a component in the third direction (the x direction), so that the angle between reconnecting magnetic field vector is less than 180 degree. This 2-component versus 3-component geometry should not be confused with 2-dimensional versus 3-dimensional geometry. As a result, there are different combinations: 2-component (or anti-parallel) reconnection in 2D or in 3D, or 3 component reconnection in 2D or in 3D. (As a side note, sometime the geometry of 3 component in 2D is called 2.5D, which could add some further confusions.) So, this manuscript is about 3-component reconnection in 3D in the experiment while 3-component reconnection in 2D in the simulation (or 2.5D), and they should not be compared as they can yield fundamentally different results. Therefore, it does not make sense to consider the simulation results reported by this manuscript in the context that the main purpose is to report experimental results.

2. In the experiment, when the guide field component was varied from zero (2-component or anti-parallel reconnection) to a finite value (3-component reconnection), the corresponding 3D geometry was also varied. When the guide field is zero ($\theta=0$), all plasma and field are co-planar, i.e. exist on the same plane. The expanding plasmas, together with the self-generated magnetic field, collide with each other in a co-planar fashion. However, when the guide field is nonzero (with nonzero θ), the T2 plane is different from the T3 plane as seen in Fig.1(b). The expanding plasma from the T2 does NOT collide with the plasma from the T3 in a co-planar fashion anymore. Around $z=0$, they still collide in such a way with a finite guide field. But away from $z=0$, two expanding plasmas, together with their self-generated magnetic field, will not collide in a co-planar fashion, and may miss each other completely. Therefore, it is not surprising to see changes in the proton reflectometry images when θ is increased from zero, but their interpretations are surely different from the 2D reconnection where plasmas collide everywhere along the z direction due to the invariance in the x direction. Any valid interpretations require taking into account the changes in the 3D geometry.

Various typos and inconsistencies are also noted but they may not be important at this stage of review.

Reviewer #3 (Remarks to the Author):

Report for

"Laboratory evidence of the halting of magnetic reconnection

by a weak guide field"

by S. Bolanos et al

General comments:

This paper reports on an experimental and numerical investigation of magnetic reconnection.

The experiment is described in detail, and it is shown that the reconnection process is profoundly modified in the presence of a mean magnetic field.

Disclaimer: my experience with magnetic reconnection does not really overlap with this work, in particular I have no real knowledge of experimental techniques.

I personally find the experimental results impressive.

I cannot follow all the details, but I see a fairly complex setup, where results have to be processed in a non-trivial manner in order to extract useful information.

Furthermore the experiments are complemented by numerical simulations, which are themselves non-trivial.

Keeping in mind the previous disclaimer, I think that the effect described is real, and worth investigating in more detail in different contexts.

Specific comments:

1. I personally find the main text quite technical.

The abstract is quite accessible, as well as the introductory paragraph and the conclusion.

The rest of the text is quite dense, and I found I needed to read some sentences several times to get their meaning.

2. I feel that I am missing some crucial details: from my reading I believe the plasmas are created by the lasers hitting a solid material, but I don't see why the protons hitting the material from the opposing side don't themselves generate plasma separately.

3. Simulations performed with the HECKLE code: it is stated that the time step is "small enough to correctly treat the high frequency whistler modes". Is this a standard stability criterion in the PIC plasma simulation literature? If not, does such a stability criterion exist? I am thinking in particular of the CFL criterion for fluid dynamics.

4. Simulations performed with the ILZ code: it is stated that "the electromagnetic field is interpolated to the 1st order". Does this mean that linear interpolation is used for the magnetic field? It is not at all clear to me if this is adequate. The magnetic field is divergence-free, and linear interpolation will not in general be able to respect this constraint. See for instance

F. Mackay, R. Marchand, and K. Kabin. "Divergence-free magnetic field interpolation and charged particle trajectory integration", *Journal of Geophysical Research*, vol 111, A06208, 2006.

Obviously, if the resolution of the grid is high enough this is not a problem, and it may be that errors due to imperfect interpolation cancel each other out on average, but I cannot tell from the available information.

5. I find the definition of "magnetic reconnection" used here to be somewhat vague. The abstract gives a rough definition of 2D magnetic reconnection. But

the conclusion mentions the solar corona where "the structure of the arches is very three-dimensional and contains many sub-substructures". Are the authors referring to turbulent reconnection in the sense of "Eyink, G. L., Lazarian, A. & Vishniac, E. T., Fast magnetic reconnection and spontaneous stochasticity. *Astrophys. J.* 743, 51 (2011)"? It is my understanding that reconnection in the presence of turbulence (one case of "many sub-substructures") is quite a different phenomenon than the 2D case mentioned in the abstract: the 2D case is strongly dependent on the nature of the plasma, whereas in the turbulent case microscopic details of the plasma become irrelevant as non-ideal effects are explosively amplified by turbulent fluctuations that are somewhat universal.

As far as I can tell the authors have something slightly different in mind, but it would probably help if the text can be clarified.

Minor comments:

6. Line 112: "such the guide field". I assume there is a missing "that".
7. Line 239: "and that magnetic field". I think there is a missing "the".
8. Lines 257 through 259: Sentence starting with "Using the simulated electric" is missing an active verb.
9. Line 307: Shouldn't "alike" be just "like"?
10. Line 309: "shape up to quite late". I don't understand what this means.
11. Line 505: "the phase velocity [...] are" --- singular/plural mismatch.

12. Line 514: I don't understand the formulation "to prevent from vacuum region".

Reviewer #1 (Remarks to the Author):

Laboratory evidence of the halting of magnetic reconnection by a weak guide field” by S. Bolanos et al., (J. Fuchs; Corresponding Author) presents the results of a laboratory experiment in which magnetic reconnection occurs at the interface of two expanding plasma cylinders produced by laser beams impinging on two Au foils. By rotating the target foils relative to one another, the orientation of the magnetic fields at the interface of the two expanding plasmas are altered, resulting in the presence of an out-of-plane “guide” magnetic field. Diagnostics measuring reconnection profiles and outflows indicate that the reconnection process is slowed and/or inhibited when the target foils are in the rotated configuration.

The title and abstract of this manuscript emphasize the role of guide field in the inhibition of reconnection. Since “guide-field” vs “anti-parallel” reconnection is a topic of significant interest, this study could potentially contribute to an understanding of the differences. I nevertheless have concerns regarding the generality of the results, as discussed below.

While the schematic description of the experiment in Figs. 1a–1c provides some sense of the reconnection geometry, it is only Supplementary Figure 14 that provides a clear (albeit idealized) representation for the nominal configuration where the rotation angle θ vanishes. I would suggest that this figure be included in the body of the manuscript.

1. My interpretation of the figure is that orange circles are cross sections of the two parallel expanding plasma cylinders. When θ does not vanish, the cylinders would no longer be parallel to one another, thereby modifying the geometry of the interaction region. Specifically, there will be inhomogeneity in the out-of-plane (x) direction absent from the $\theta = 0$ case. The presence of this inhomogeneity occurs only for those cases where the guide field does not vanish thereby complicating the role of the guide field on the inhibition of reconnection. Indeed, this change in geometry may be more important than the change in topology referred to in the Abstract (i.e., from 2D to 3D). Note that the importance of a transition to 3D topology is questionable since even in the absence of an initial guide field, a significant out-of-plane B_x develops spontaneously in the anti-parallel configuration as evidenced by the quadrupolar Hall magnetic field pattern in the simulation (Fig. 5b).

We thank the reviewer for raising this issue, because it is indeed central to validate our approach to the investigation of the influence of a guide-field. We now realize we have certainly not clarified it enough.

- a. The major point that needs clarification is the type of magnetic reconnection (MR) configuration we investigated in the experiment. Two major types of MR

are shown in Fig. A below, namely the so-called *X-point* configuration and what is known as the *elongated* MR. Fig. A.a shows the “X-point’ configuration where two plasmas having a high curvature of their B-field lines meet at A SINGLE POINT. In this configuration, the B-fields on the right and left of that X-point do NOT interact. These parts of the plasma feed the contact point where MR takes place, but they do not directly participate to MR in itself. This is the configuration we use in the experiment, precisely because we use two B-fields which are disks, and where there is only a single contact point between the two structures.

The second canonical configuration is the one illustrated in Fig. A.b where two elongated (e.g. very elliptical or even parallel, like in the Earth’s magnetotail) plasma and B-field structures interact all along a long current sheet. In this case, obviously, since MR is taking place all along the elongated structure, tilting one structure with respect to the other would affect the way MR would take place as the configuration would be modified. This is not the configuration used in our experiment.

We apologize for that lack of clarity on this central issue. We chose this particular geometry, namely the X-point MR, to investigate the influence of the guide field. Therefore, as our configuration is an X-point one, tilting one plasma with respect to the other one does not disturb MR in taking place, since it happens only at one point. But doing so introduces a guide-field (GF), the influence of which we can then evaluate.

The two main canonical magnetic reconnection configurations:

Fig.A: The two types of canonical magnetic reconnection configurations.

Now, as mentioned in the manuscript, the point is that imposing this GF everywhere around the X-point makes it harder for the spontaneous Hall quadrupolar magnetic field to establish itself. Where the polarity of the Hall field is along the GF, it would help the Hall field. However, in the other quadrants where the Hall field is opposite to the GF, the latter needs to be quenched in

order for the Hall field to grow, which explains the slowing down of the triggering of MR.

- b. The second point to clarify is the actual topology of the B-fields we employed. The two plasmas that are put in contact are actually not cylinders, but rather think disks constrained along the target surface. To visualize this, we present in Fig.B below a map of the B-field compressed along the target surface, following laser irradiation of a solid, in conditions similar to that of the experiment. This topology of the B-field has been validated by proton probing measurements and has been presented in detail in (Lancia *et al.*, 2014).

*Fig.B: 2D map of the magnetic field produced following the laser-irradiation of a solid target in LULI2000 conditions, i.e. in conditions similar to that of the experiment which is reported in the present paper. The map is obtained from an hydro-rad simulation (FCI2 code), performed in an axi-symmetric geometry. Vertical is the radial axis, and horizontal is the longitudinal axis, aligned with the target normal. We can see that the B-field is compressed in a thin disk along the critical density layer (red line). Reprinted (Fig.3.a) with permission from [L. Lancia *et al.*, *Physical Review Letters*, Volume 113, p. 235001, 2014].*

Copyright (2014) by the American Physical Society.

To clarify and emphasize these two very important points, we have modified the title as follows (added text is in red): “Laboratory evidence of **hampered X-point** magnetic reconnection by a weak guide field”. We have also revised Fig.1 of the paper to try to clarify the setup we use, and we have modified the paper at lines 31, 56, 67, 85, 91 and 151.

2. Note that it is typical in simulation studies addressing the role of an initial guide field for the magnitude of the reversing magnetic field to be held constant. In the present study, it appears instead to be the total magnetic field that is held constant. Therefore, at least some of the inhibition of reconnection in the presence of the guide field is presumably due to the reduction in the strength of the in-plane reversing magnetic field by a factor of $\cos(\theta/2)$. This could potentially account for the delay in the timing of the spike in the reconnection rate in the simulations (e.g., Fig. 5a vs Fig. 5c).

We agree with the reviewer that, unlike most numerical studies on the influence of a guide field (GF), where the in-plane magnetic field is kept constant, we keep the overall magnetic field (in-plane and guide) constant. True, this is due to the constraint of the experiment. However, we believe that this is also a more interesting setup with respect to some natural MR events. Take for example a solar arch that is subject to twisting. Indeed, the twisted branch of the arch will keep its total magnetic field constant, meaning that the projected magnetic field will decrease, as the expense of the growing guide-field. This is different from keeping a constant projected in-plane magnetic field, while varying the GF, which would be somewhat artificial.

Now, the reviewer wonders naturally what is the impact of reducing the strength of the projected in-plane magnetic field, namely does it slow down significantly reconnection? To answer this question, we first note, as now added in the paper at line 103:

“[The] factor [$\cos(\theta/2)$] of the in-plane magnetic field strength, when tilting the targets] is 0.92 in the extreme case used here of $B_{GF}/B_0 = 0.41$. We first note that the shot-to-shot variation of the laser energy in the experiment (standard deviation / mean = 0.93) is close to 0.92, and that despite such variation the shots are very reproducible in terms of temporal evolution of the reconnection. In particular, we have higher energyshots in the guide field configuration than in the coplanar one, and they still show the same delayed guide-field reconnection.”

Second, we ran a simulation in a coplanar configuration, but where the magnetic strength was reduced by a factor $\cos(\theta/2)$ in order to reproduce the reduction in strength of the in-plane magnetic field when we tilt one target compared to the other. In Fig.C below, the evolution of the reconnection rate is represented for this simulation as well as for that of the simulation in the coplanar case.

Fig.C shows clearly that the magnetic reconnection rate is very little affected by the asymptotic strength of the magnetic field. This is not surprising given that the strength of the in-plane magnetic field is only slightly lower, by a factor of 0.92. Such factor is the one corresponding to a guide field of $B_{GF}/B_{YZ} = 0.41$, i.e. to a maximum tilt of 45°

between the two targets. This additional simulation, which is now added to the Supp. Info. of the paper (Supp. note 7), confirms that the strong time-lag we observe both in the experiment and the simulations when we impose even a small GF is not caused by the decrease in the in-plane magnetic field strength. It is rather due to the fact that the GF destabilizes the quadrupolar Hall field which is much more difficult to establish when a single polarity GF is present all around the MR X-point.

Fig. C: Simulations, using the HECKLE hybrid code, of the temporal evolution of the reconnected magnetic flux (top panel) and of the reconnection rate (bottom panel) for two coplanar simulations. The difference between the two simulations, which use the same setup as the simulations performed in the main text of our paper is only that the strength of the initial magnetic field is varied. In the first simulation, the strength of the magnetic field is $B = B_0$ (blue curve) whereas in the second simulation, the strength of the magnetic field is $B = 0.92 \times B_0$ (red dotted curve).

This is now clarified in the paper at line 103, and also in the Supplementary Note 7.

3. Since the simulations appear to be two-dimensional, they would not be able to address the out-of-plane inhomogeneity in the experimental setup previously discussed. Such 3D simulation studies might be able to account for additional dependencies on the angle θ .

First, we would like to underline that the evidence put forward in the paper regarding the strong slowing down of MR on a X-point magnetic reconnection event is based on experimental data, irrespective of any numerical simulations.

Second, when tilting one target with respect to another an inhomogeneous guide-field is then resulting. As discussed in the previous point (#2), the modification of the in-plane strength of the magnetic field induced by the tilt has indeed no impact on the evolution of MR.

So, overall, we believe that simulations, which are 2D with respect to the spatial distribution of the plasma, but which are 3D in terms of fields, as used in the paper, are adequate to investigate the destabilization of the MR event induced by imposing a guide-field, as we do in the experiment.

This is now clarified in the paper as follows (line 134):

“The appropriate numerical tool to simulate such a situation has to address the well-known duality between a large scale problem (to properly treat the inflow and outflow regions without arbitrary boundary conditions) and its microphysics description at least at the ion scale. The two-dimensional version (for the positions, although the fields and velocities are treated in three dimensions) of the hybrid HECKLE code (see Methods) appears in this respect to be the best compromise, in order to support the interpretation of the laboratory experiments. Note that the kinetic behavior of the electrons is mimicked in the simulations¹¹ through the integration of an hyper-resistive term in the generalized Ohm's law.”

4. As a final note regarding the simulations, it is known that electron kinetics (such as off-diagonal terms in the pressure tensor) can influence the reconnection rate. Such effects are absent in the hybrid simulations described here.

The reviewer is right in raising this issue, which was not clarified enough in the first version of the paper. Reconnection is a multi-scale phenomenon, that is to say that small scales (ions and electrons) as well as large scales (MHD) play a role in the dynamics of the reconnection event. The kinetics of the electrons influence the phenomenon as well as does the plasma on the MHD scale. In addition, the coupling between large-scale and small-scale parameters remains to this day one of the main problems linked to numerically investigating magnetic reconnection. However, to be able to simulate plasma on the MHD scale while describing kinetically the electrons, it is necessary to have an enormous computational capacity, which is still out of reach, and will stay so in the foreseeable future.

A large number of studies (Hesse *et al.*, 1999; Fox, Bhattacharjee and Germaschewski, 2011; Aunai, Hesse, Black, *et al.*, 2013; Joglekar *et al.*, 2014) have decided to focus on the small scales, that is, the kinetics of the electrons, but this means that they do this at the expense of describing the MHD scale. Indeed, in order to be able to follow the electrons over a duration similar to that of the duration of the laser pulse (5 ns), it is necessary to simulate electrons for 42,000 gyroperiods. This can be done in practice only by limiting the simulation over a very small region in space, typically over a few hundred electron inertia lengths. Thus, they do this at the expense of describing the overall dynamics of the expanding plasma.

For us, we made the choice of simulating the expanding plasma plumes rather than the kinetic behaviour of the electrons. We did this because we believe that in a case of forced reconnection driven by two fastly convergent inflows, as is the case in our experiment, the large scales are the most crucial to describe. This is due to the fact that in a forced configuration, the inner plasma pressure within the expanding plasmas is higher than the outside pressure, precisely driving the expansion, and the dynamics

of the encounter of the inflows. Hence, describing the overall plasmas is important to describe how the inflows meet. A description limited to a small region centered around the X-point and which would not integrate the hot center of the plasma bubbles would miss that point.

Practically, this means that we chose to describe both the MHD and the ion kinetic scales. This is done by using a hybrid-PIC code, where ions are treated like macroparticles and electrons like a massless fluid. In order to close the system, an isothermal hypothesis was imposed. This assumption therefore leads to neglect the agyrotropic terms of the electronic pressure tensor.

Despite this, i.e. the practical impossibility to describe at the same time the MHD scale and the electrons kinetics, some tricks can be played to emulate the physics of electron kinetics at MHD levels. For example, in order to mimic the behavior of these agyrotropic terms, Hesse et al. 2011 proposed adding a term of hyperviscosity to Ohm's generalized law. This term seems to correctly restore the kinetic behavior of electrons (agyrotropic terms) in the context of magnetic reconnection (Aunai, Hesse, Zenitani, *et al.*, 2013).

Through this direct comparison between hybrid and fully kinetic simulations, the authors demonstrated that the reconnecting systems behave in a similar way. The force balance presents identical profiles and also for energy transfer. In conclusion, taking a hybrid model to describe the entire plasma plumes and the diffusion region is the best choice in regard to the computational cost.

This has been clarified in the paper, at lines 140 and 549.

While I do not question the results of the experiments presented in this manuscript, I cannot accept—without further justification—the conclusion implicit in the title that it is the presence of the weak guide field alone that controls the rate and character of the reconnection.

We agree with the reviewer that we did not clearly enough expose that our setup, being one of an X-point MR event, is not influenced per se in tilting the target, which however brings the benefit of adding a guide-field to the MR event. With this clarification, we hope that the reviewer is now convinced that our setup is “clean”, meaning that it indeed allows to elucidate the influence of a guide-field on MR, without otherwise disturbing MR. And that our results indeed show that it is clearly the guide- field that slows down the dynamics of MR, the most plausible explanation behind it being the destabilization of the Hall quadrupolar field.

Furthermore, the nature of reconnection studied is an example of driven reconnection where the plasma regions in which the reversing magnetic fields are embedded are dynamically forced together. Many reconnection scenarios

involve plasmas that are juxtaposed, but otherwise initially stationary so that the reconnection is initialized by means of an instability such as tearing. Caution is needed when applying the conclusions of a particular study to a more general scenario.

We agree with the reviewer that our findings cannot be generalized to all possible MR configurations, in particular to those involving an elongated MR zone (see Fig.A.b) where the tearing instability seem to play a major role in controlling the triggering of MR. We have now clarified throughout the text, lines 85, 96 and 151, that our configuration is that of an X-point, and that our findings apply to that configuration.

This is now also clearly emphasized in the modified title:

“Laboratory evidence of **hampered X-point** magnetic reconnection by a weak guide field”

In consideration of the wide variety of reconnection regimes, please also note that while the references cited in this manuscript do point to representative examples of prior theoretical, numerical, and experimental studies, they do not necessarily convey the vast number and breadth of such studies—including investigations of guide-field dependence.

We agree with the reviewer that there are a very large number of investigations on the topic of MR, including those on the role of a guide-field. Given the constraints of the number of references that can be used in the paper, we tried to include representative papers as references, but we might have missed some that the reviewer could see more important than the ones we cited. We would be happy if the reviewer would indicate some specific and important references that we would have missed.

In light of the above considerations, I believe that the experiment reported in this study is deserving of publication, but only if the conclusions are couched in a less general context. For example, a title such as “Laboratory evidence of the halting of magnetic reconnection in the interface between non-parallel expanding plasma cylinders with embedded azimuthal magnetic fields” might more accurately describe the current experiment. Of course, doing so might also suggest that a more specialized journal than Nature Communications would be preferable as a publication venue.

We thank the reviewer for having highlighted the lack of clarity, from our part, in the configuration used in our experiment, i.e. a X-point configuration that is insensitive to a tilt of one target with respect to the other. A strong advantage of such X-point configuration is that it is not influenced by other factors, e.g. like the tearing instability that plays a role in elongated MR configuration. Hence, we believe that the X-point

configuration we used is actually very adequate to solely evaluate the effect of adding a guide-field, without otherwise influencing the MR configuration.

In the end, we believe that the demonstration we bring of the surprisingly strong influence of a guide-field on the dynamics of a MR event, unspoiled by other factors, could be of interest to the broad community interested in MR and magnetic field dynamics in plasmas.

Thus, we hope that the clarification we brought on this point to the paper are satisfactory and that the reviewer will find the revised version of the paper suitable for publication.

Reviewer #2 (Remarks to the Author):

Magnetic reconnection is an important fundamental plasma process occurring in many astrophysical and laboratory fusion plasmas. There exist a large number of unanswered major problems such as when and how does magnetic reconnection initiate and at what speed does it proceed under various magnetic geometries. This manuscript describes laboratory experiments and numerical simulations designed to address one particular problem on magnetic geometries: dependence of reconnection process on the guide field strength.

However, I have the following two major issues on the main conclusions of this manuscript which prevent me from recommending it be accepted for publication on Nature Communications:

- 1. The experimental setup is a truly 3D geometry which does not have an invariant direction to qualify as a 2D process. The assumed reconnection takes place in the reconnection plan (the y-z plan) and is very localized in the third direction (the x direction). This is in contrast to the simulation setup which uses a truly 2D geometry where the out-of-the-plane direction (the x direction) is the invariant direct, i.e., the reconnection process takes place in an exactly same fashion at all x locations from negative infinity to positive infinity. The commonality between experiment and simulation is only about the third component of magnetic field vector, or the guide field component (the x component). Without a guide field, the reconnecting magnetic field vector remains in the reconnection plane (the y-z plan) while with a finite guide field, the magnetic field vector has a component in the third direction (the x direction), so that the angle between reconnecting magnetic field vector is less than 180 degree. This 2-component versus 3-component geometry should not be confused with 2-dimensional versus 3-dimensional geometry. As a result,**

there are different combinations: 2-component (or anti-parallel) reconnection in 2D or in 3D, or 3 component reconnection in 2D or in 3D. (As a side note, sometime the geometry of 3 component in 2D is called 2.5D, which could add some further confusions.) So, this manuscript is about 3-component reconnection in 3D in the experiment while 3-component reconnection in 2D in the simulation (or 2.5D), and they should not be compared as they can yield fundamentally different results. Therefore, it does not make sense to consider the simulation results reported by this manuscript in the context that the main purpose is to report experimental results.

We agree with the reviewer that there was a lack of clarity, from our part, in the wording about dimensions and geometry. This confusion resulted in a lack of clarity regarding what the experiment allowed us to infer, and how valid our results were. All this discussion has common points to the ones raised by reviewer #1, but we will detail them again here (see also the answers to the points raised by reviewer #1). But we also want first to stress that the main conclusion we can derive in the paper regarding the influence of a guide field (GF) on magnetic reconnection (MR) is obtained from experiments. The simulations are here to help us confirm the intuition that the GF can slow down MR through the destabilization of the quadrupolar Hall field. Nonetheless, the main point that the introduction of a GF indeed strongly slows down MR can be purely derived from the experimental results. This is essential.

This said, we want now to clarify this issue of dimensions and geometry. The main point is that the configuration we use is an X-point configuration of MR, i.e. one where the contact between the two anti-parallel magnetic structures is realized only at a SINGLE POINT in space. We have fields in 3D (because the encountering fields are 2D, but that a spontaneous 3D Hall will grow) that meet at a single point (see also answer #1 to reviewer 1). This is the essential point.

It is essential because when we tilt one target with respect to the other, we truly add a GF with a single polarity everywhere around the X-point, without otherwise changing that X-point configuration. Thus the chosen experimental configuration allows us to evaluate quantitatively the influence of a GF, without any other parasitic effects entering into the picture.

To clarify all this, we have rephrased several sections of the paper at lines 93, 97, 151.

2. In the experiment, when the guide field component was varied from zero (2-component or anti-parallel reconnection) to a finite value (3-component reconnection), the corresponding 3D geometry was also varied. When the guide field is zero ($\theta=0$), all plasma and field are co-planar, i.e. exist on the same plane. The expanding plasmas, together with the self-generated

magnetic field, collide with each other in a co-planar fashion. However, when the guide field is nonzero (with nonzero theta), the T2 plane is different from the T3 plane as seen in Fig.1(b). The expanding plasma from the T2 does NOT collide with the plasma from the T3 in a co-planar fashion anymore. Around $z=0$, they still collide in such a way with a finite guide field. But away from $z=0$, two expanding plasmas, together with their self-generated magnetic field, will not collide in a co-planar fashion, and may miss each other completely. Therefore, it is not surprising to see changes in the proton reflectometry images when theta is increased from zero, but their interpretations are surely different from the 2D reconnection where plasmas collide everywhere along the z direction due to the invariance in the x direction. Any valid interpretations require taking into account the changes in the 3D geometry.

Again, we apologize for the confusion on this important issue, which is linked to the one discussed in the previous point raised by the reviewer. The main point here is that the configuration of the reconnection is that of an X-point, i.e. where reconnection really takes place at a single point, and where the plasmas on both sides do not interact, but merely play the role of feeding the X-point with inflows (see Fig.A above and the answer to reviewer 1).

This is fundamentally why it is not a problem to have the two disks of magnetized plasmas not in the same plane. As long as they keep a point of contact, which is the X-point where reconnection occurs, reconnection can occur in the same way (apart of course from the addition of the guide-field, which is the point of the present study).

We stress that the fact that we are in a X-point reconnection geometry is not solely inferred from the simulations as shown in Fig.2 and Fig.5 of the paper, but is also a conclusion that can be robustly inferred from looking at the parameters describing the magnetic reconnection events provoked here. Indeed, the Lundqvist number is $S = 100$. Such a configuration of reconnection leaves no doubt about reconnection taking place at a single X point, see (Ji and Daughton, 2011; Daughton and Roytershteyn, 2012). Even in the coplanar case where the two plasmas are in the same plane, magnetic tension prevents the magnetic loops from collapsing onto each other and to form an elongated area where reconnection would occur - as indeed observed in the simulations. In short, whether or not there is a guide field, magnetic reconnection occurs at a single point for the parameters at play in the experiment.

Various typos and inconsistencies are also noted but they may not be important at this stage of review.

Reviewer #3 (Remarks to the Author):

Report for "Laboratory evidence of the halting of magnetic reconnection by a weak guide field" by S. Bolanos et al

General comments:

This paper reports on an experimental and numerical investigation of magnetic reconnection. The experiment is described in detail, and it is shown that the reconnection process is profoundly modified in the presence of a mean magnetic field.

Disclaimer: my experience with magnetic reconnection does not really overlap with this work, in particular I have no real knowledge of experimental techniques.

I personally find the experimental results impressive.

I cannot follow all the details, but I see a fairly complex setup, where results have to be processed in a non-trivial manner in order to extract useful information.

Furthermore the experiments are complemented by numerical simulations, which are themselves non-trivial. Keeping in mind the previous disclaimer, I think that the effect described is real, and worth investigating in more detail in different contexts.

Specific comments:

1. I personally find the main text quite technical.

The abstract is quite accessible, as well as the introductory paragraph and the conclusion. The rest of the text is quite dense, and I found I needed to read some sentences several times to get their meaning.

We thank the reviewer for his/her supportive comments. We realize also, reading the comments of reviewers 1 and 2 that we were not clear enough on the geometry and configuration we used, namely that of an X-point type reconnection, which is a critical fact here since it validates our approach to investigate guide-field reconnection. As a result, we have rewritten in depth all the presentation of the experiment and all the sections detailing how the magnetic field topology is inferred (see the text in red in the revised version of the paper). We hope that the reviewer finds the revised text more clear.

2. I feel that I am missing some crucial details: from my reading I believe the plasmas are created by the lasers hitting a solid material, but I don't see why the protons hitting the material from the opposing side don't themselves generate plasma separately.

The reviewer raises a good point which we realize we did not clarify enough, as accustomed we are of laser-plasma experiments. The main point here is that the protons traversing the targets T2 and T3 (see Fig.1 of the main text), where the magnetized plasmas are generated, have a very low density when they traverse these targets. Indeed, they have been expanding (with a cone of half-angle $\sim 20^\circ$) from the proton source target T1. Hence, their density is completely negligible compared to that of the targets T2 and T3, and moreover their energy density is very small (37 MPa) compared to that of the plasma generated by the lasers (600 GPa). Hence, their passing through the targets T2 and T3 does not change the characteristics of the laser-driven plasmas on these targets.

To clarify all this, we have added the following text in the Methods section, line 497:

“Note that the energy density of a proton beamlet is negligible with respect of the energy density in the plasma plume driven by the nanosecond beams (L2 and L3 in the cartoons of Fig.1). Hence, the plasma plumes are not affected by the presence of the propagating protons.”

3. Simulations performed with the HECKLE code: it is stated that the time step is "small enough to correctly treat the high frequency whistler modes". Is this a standard stability criterion in the PIC plasma simulation literature? If not, does such a stability criterion exist? I am thinking in particular of the CFL criterion for fluid dynamics.

Indeed, the time-step is 1×10^{-14} s in order to satisfy the CFL conditions. This is now mentioned clearly in the Methods section, at line 570.

4. Simulations performed with the ILZ code: it is stated that "the electromagnetic field is interpolated to the 1st order". Does this mean that linear interpolation is used for the magnetic field? It is not at all clear to me if this is adequate. The magnetic field is divergence-free, and linear interpolation will not in general be able to respect this constraint. See for instance

F. Mackay, R. Marchand, and K. Kabin. "Divergence-free magnetic field interpolation and charged particle trajectory integration", Journal of Geophysical Research, vol 111, A06208, 2006.

Obviously, if the resolution of the grid is high enough this is not a problem, and it may be that errors due to imperfect interpolation cancel each other out on average, but I cannot tell from the available information.

The article by Mackay et al. presents a method of calculating the magnetic field by symplectic integration, which improves the conservation of the first adiabatic invariant, the magnetization of particles. However, the ILZ code simulates the propagation of the proton beam probing the plasma. In practice, these probe protons are highly demagnetized. Typically, we look at the deflections of the protons having an energy of the order of 10 MeV, i.e. that the radius of Larmor is of the order of 23 mm compared to the sub-millimeter plasma.

The ILZ code does not calculate the evolution of the electromagnetic field and the plasma. Using this code, we are assuming that the plasma is static. 10-MeV protons take 2.3 ps to pass through 100 μm of magnetized plasma (the characteristic thickness of the magnetic disk, see Supplementary Figure 1). The propagation time is, therefore, much less than the characteristic Alfvén time and the scale time of the magnetic reconnection event (sub ns). For this reason, a static hypothesis is justified, and looking at the conservation of the first adiabatic invariant is out of the scope.

5. I find the definition of "magnetic reconnection" used here to be somewhat vague. The abstract gives a rough definition of 2D magnetic reconnection. But the conclusion mentions the solar corona where "the structure of the arches is very three-dimensional and contains many sub-substructures". Are the authors referring to turbulent reconnection in the sense of "Eyink, G. L., Lazarian, A. & Vishniac, E. T., Fast magnetic reconnection and spontaneous stochasticity. *Astrophys. J.* 743, 51 (2011)"? It is my understanding that reconnection in the presence of turbulence (one case of "many sub-substructures") is quite a different phenomenon than the 2D case mentioned in the abstract: the 2D case is strongly dependent on the nature of the plasma, whereas in the turbulent case microscopic details of the plasma become irrelevant as non-ideal effects are explosively amplified by turbulent fluctuations that are somewhat universal.

As far as I can tell the authors have something slightly different in mind, but it would probably help if the text can be clarified.

We agree with these remarks, and our purpose was not to slide on this side of magnetic reconnection. Modelling is an approach that allows to address the importance of parameters/features by isolating them in a simplified context. We just wanted to emphasize that remote-sensing observations of magnetic reconnection in solar arches do not allow us to infer any details (to our knowledge) associated with the microphysics processes. While 3D, the experimental set-up we present has the advantage to be stripped from such small-scales fluctuations, hence allowing us to address how the reconnection rate depends on the guide field.

Minor comments:

6. Line 112: "such the guide field". I assume there is a missing "that".

This sentence has been rephrased.

7. Line 239: "and that magnetic field". I think there is a missing "the".

Here, "that" is used as a demonstrative adjective. Thus, using "the" would be redundant.

8. Lines 257 through 259: Sentence starting with "Using the simulated electric" is missing an active verb.

This sentence has been rephrased.

9. Line 307: Shouldn't "alike" be just "like"?

This sentence has been rephrased.

10. Line 309: "shape up to quite late". I don't understand what this means.

This sentence has been rephrased.

11. Line 505: "the phase velocity [...] are" --- singular/plural mismatch.

We thank the reviewer for spotting these. All these mistakes have now been corrected (or the corresponding sentences have been rephrased).

12. Line 514: I don't understand the formulation "to prevent from vacuum region".

The sentence has been rephrased: "to prevent the formation of a vacuum region".

References

Aunai, N., Hesse, M., Zenitani, S., *et al.* (2013) 'Comparison between hybrid and fully kinetic models of asymmetric magnetic reconnection: Coplanar and guide field configurations', *Physics of Plasmas*, 20(2), p. 022902. doi: 10.1063/1.4792250.

Aunai, N., Hesse, M., Black, C., *et al.* (2013) 'Influence of the dissipation mechanism on collisionless magnetic reconnection in symmetric and asymmetric current layers', *Physics of Plasmas*, 20(4), p. 042901. doi: 10.1063/1.4795727.

Daughton, W. and Roytershteyn, V. (2012) 'Emerging parameter space map of magnetic reconnection in collisional and kinetic regimes', *Space Science Reviews*,

172(1–4), pp. 271–282. doi: 10.1007/s11214-011-9766-z.

Fox, W., Bhattacharjee, A. and Germaschewski, K. (2011) 'Fast Magnetic Reconnection in Laser-Produced Plasma Bubbles', *Physical Review Letters*, 106(21), p. 215003. doi: 10.1103/PhysRevLett.106.215003.

Hesse, M. *et al.* (1999) 'The diffusion region in collisionless magnetic reconnection', *Physics of Plasmas*, 6(5), pp. 1781–1795. doi: 10.1063/1.873436.

Ji, H. and Daughton, W. (2011) 'Phase diagram for magnetic reconnection in heliophysical, astrophysical, and laboratory plasmas', *Physics of Plasmas*, 18(11), p. 111207. doi: 10.1063/1.3647505.

Joglekar, A. S. *et al.* (2014) 'Magnetic Reconnection in Plasma under Inertial Confinement Fusion Conditions Driven by Heat Flux Effects in Ohm's Law', *Physical Review Letters*, 112(10), p. 105004. doi: 10.1103/PhysRevLett.112.105004.

Lancia, L. *et al.* (2014) 'Topology of Megagauss Magnetic Fields and of Heat-Carrying Electrons Produced in a High-Power Laser-Solid Interaction', *Physical Review Letters*, 113(23), p. 235001. doi: 10.1103/PhysRevLett.113.235001.

Reviewers' comments:

Reviewer #1 (Remarks to the Author):

Please see uploaded PDF file, including one embedded figure.

I appreciate the effort the authors have made in revising Manuscript NCOMMS-19-26005A in response to previous reviewers' comments—including the change of title to “Laboratory evidence of hampered X-point magnetic reconnection by a weak guide field.” Nevertheless, some of my initial reservations remain.

The crux of my concern relates to the (new) text on p. 2 (line 96): “We stress that, since we are in a X-point configuration, the plasmas and magnetic fields on both sides of the reconnection point interact locally and only at the X-point site. . .” Even when reconnection is restricted to a single X-point, it is an overstatement to claim that reconnection is a strictly local phenomenon. One cannot isolate the local magnetic field topology from the plasma geometry in the vicinity of the X-point. Similarly, I cannot agree with the following statement in the rebuttal letter: “In this configuration, the B-fields on the right and left of that X-point do NOT interact. These parts of the plasma feed the contact point where MR takes place, but they do not directly participate to MR in itself.” Magnetic reconnection (MR) is not simply the local change in magnetic topology, but is a plasma process that involves both fields and particles, including flows and gradients. Indeed, the authors appear to be in agreement with this point when they say the following in their rebuttal letter: “Reconnection is a multi-scale phenomenon, that is to say that small scales (ions and electrons) as well as large scales (MHD) play a role in the dynamics of the reconnection event. It is on these larger MHD scales that the geometry is particularly relevant. Please note that the distinction between X-point and *elongated* MR—as discussed in the rebuttal letter—was never a point of confusion regarding the original presentation. The non-locality of the reconnection process holds for both cases.

The addition of the face-on view (c) and 3D view (d) in the schematic of Fig. 1 do add some additional context to the geometry of the experiment. However, the side view (a) is still somewhat misleading. What is shown is not the configuration as viewed side-on from a distance, but rather a slice through the configuration in the x - y plane at $z = 0$. If the slice were made at some other (small) value of z , the orange rectangles (foils T2 and T3) would be offset horizontally from one another.

To better illustrate my concerns, I have prepared a figure (below) that contains what I believe to be a more global depiction of the 3D geometry of the interaction region, based upon the information in panel (d) of Fig. 1 in the manuscript. While the top view is very similar to panel (a) of Fig. 1 in the manuscript, the side view and 3D view show more clearly the geometry of the interaction region when the two foils are rotated relative to one another. While the rotation of the foils *does* result in there being an effective guide field at the point of contact between the two outermost rings, it is *not* self-evident that the simultaneous change in local geometry at the contact point will not influence the reconnection dynamics. Thus, ascribing the dependence on angle ϑ to the magnitude of the guide field *alone* is an oversimplification. Based on the preceding considerations, I find the following sentence of the Abstract (line 30) to be misleading: “Here we test, in the laboratory and using the canonical “X-point” configuration, where the two magnetized plasmas meet at a single point, the effect of imposing an out-of-plane (guide) magnetic field which breaks the initial symmetry of the system.” It would be more appropriate to say that the guide field *arises* as a consequence of the geometry of the setup than to say that it is *imposed*.

I wish to reiterate that I consider the authors to have provided a thorough and accurate analysis of their experimental results. My reservations lie not with the analysis, but with the interpretation—especially with the attribution of differences due to variation of the angle ϑ to the change in guide field alone. In other words, I remain unconvinced by the authors' assertion (from the rebuttal letter) that “In the end, we believe that the demonstration we bring of the surprisingly strong influence of a guide-field on the dynamics of a MR event, *unspoiled by other factors*, could be of interest to the broad community interested in MR and magnetic field dynamics in plasmas.” [emphasis added.] Because reconnection depends on the characteristics of the plasma beyond the immediate vicinity of the X-point, variation in “other factors” must be taken into account.

Regarding the role of simulations in this manuscript, I accept the authors' assertion that “First, we would like to underline that the evidence put forward in the paper regarding the strong slowing down of MR on a X-point magnetic reconnection event is based on experimental data, irrespective of any numerical simulations.” However, it must be kept in mind that a 2D-3V simulation is, in effect, infinite but uniform in the third spatial dimension, which is not a particularly good representation of a plasma confined to a pair of disks (irrespective of the angle ϑ). The fact that the 2D simulations show an inhibitory effect of adding a guide field is significant. Indeed, I certainly do not mean to suggest that a guide field cannot have such an effect.

Figure 1: Schematic representation of experimental geometry using 3D surfaces. The red arrows indicate the direction of \mathbf{B} .

It is the degree of the effect that is at question (see next paragraph). Again, my main point is that other contributing factors cannot be ignored when interpreting the experimental results.

As the authors point out in their introduction, past simulation studies of guide-field reconnection have yielded somewhat contradictory results. Since not all of these studies involve reconnection at an X-point, I will focus on Refs. [16] and [20], both of which *do* simulate X-point reconnection. While Ref. [16] only considers guide-field ratios $B_g/B_0 = 1$ and 5, Ref. [20] treats guide-fields ratios $B_g/B_0 < 1$ as well. Reference [16] is nevertheless significant because it extends the “GEM Reconnection Challenge” study so as to include a guide field [“Geospace Environmental Modeling (GEM) magnetic reconnection challenge,” J. Birn et al., *J. Geophys. Res.*, **106**, (2001) doi:10.1029/1999JA900449]. Both studies [16] and [20] find X-point reconnection to be somewhat inhibited by the presence of a guide field, but neither is consistent with the following statement in the Abstract of this manuscript: “A strong slowing down and even halting of reconnection process is observed, even for a weak guide-field.” If the claim that the experimental results presented in this manuscript are valid for X-point reconnection in general, then one must conclude that the aforementioned simulation studies are therefore invalid. Alternatively, one must assume that there is some fundamental difference in the dynamics and geometry of the respective reconnection scenarios that accounts for the differences. I believe the *latter* interpretation is the correct one. In either case, the final sentence of the abstract, (“These observations stress the importance of taking into account guide-field effects in reconnection taking place in natural and laboratory environments.”) is a point well taken, albeit one that has been long recognized as well. In this regard, it should be noted that the presence of a guide field plays a more significant role in suppressing *asymmetric* reconnection [“Diamagnetic suppression of component magnetic reconnection at the magnetopause,” M. Swisdak et al., *J. Geophys. Res.*, **108**, (2003), <https://doi.org/10.1029/2002JA009726>], for reasons other than those discussed in this manuscript.

In my original report, I suggested an alternative title that would more accurately describe the experiment. As the authors have pointed out, the interaction is between disks rather than cylinders. I appreciate this clarification. However, after taking that clarification into account, I still believe that a revised title along the lines of the following would be a more accurate description: “Laboratory evidence of the hampered X-point magnetic reconnection at the interface between non-parallel expanding plasma disks with embedded azimuthal magnetic fields.” To some degree, the fact that that plasma is localized to adjacent disks makes it harder to envision a physical scenario (e.g., in a space or astrophysical context) where such a geometry would provide a natural model. (Adjacent cylinders, on the other hand, suggest scenarios such as interacting flux ropes in the solar corona, for example.) The authors are, of course, welcome to provide illustrative examples of naturally occurring regimes where their experimental geometry would be a good approximation.

In summary, if the experiment described in this manuscript were in fact—as the authors claim—representative of X-point reconnection in general, then this work would certainly be appropriate for publication in *Nature Communications*. However, as detailed in my report, I believe the claim of *general* applicability is overstated. Provided that the claims implying the generality of the experimental results are sufficiently softened, I would be happy to recommend publication of this study. Before concluding that *Nature Communications* is (or is not) the appropriate venue for publication, I wish to afford the authors the opportunity to make the necessary modifications and to then argue their case in this regard.

Reviewer #2 (Remarks to the Author):

Report on the revised NCOMMS-19-26005A, "Laboratory evidence of the halting magnetic reconnection by a weak guide field", by S. Bolanos et al.

The authors have made attempts to address some concerns that I have raised in my previous report but I do not think they were successful. The point is not about the shape of the diffusion region as shown in Fig.A in their reply; it is about whether the reconnection takes place in 2D or 3D. In 2D, there is an invariant direction along which everything occurs in exactly the same way while in 3D, there does not exist such a direction. I have explained the differences in the points #1 and #2 in my previous report that I will not repeat here. Therefore, reconnection occurs in fundamentally different ways in 2D vs 3D, and the 2D understanding cannot be applied to 3D without justifications. Such justifications should not be easy as I explained in detail in point #2 in my previous report. The experimental setup is highly 3D and reconnection occurs only at one point in the 3D space, with a lot of freedom for plasma to flow around about it. In order to develop a correct physics interpretation of the experimental results, the 3D geometry needs to be taken into account. Therefore, I cannot recommend the acceptance of this manuscript in the present form for publications in Nature Communications.

Reviewer #3 (Remarks to the Author):

I would like to thank the authors for their explanations.

I believe the point I raised have been addressed in a reasonable manner.

There is one typo that stood out: line 365, the words "loosing efficiency" should probably be "losing efficiency".

But otherwise I personally think the paper is ready to be published.

While not fully within my area of expertise and quite technical, I believe the draft will be of interest to a fairly wide plasma physics audience.

Answer to the last round of reviewers' comments

The reviewer's comments are in **blue**; our answers are in light black. All the modified text in the new version of the paper is in **purple**.

Reviewer #1 (Remarks to the Author):

I appreciate the effort the authors have made in revising Manuscript NCOMMS-19-26005A in response to previous reviewers' comments—including the change of title to “Laboratory evidence of hampered X-point magnetic reconnection by a weak guide field.” Nevertheless, some of my initial reservations remain.

The crux of my concern relates to the (new) text on p. 2 (line 96): “We stress that, since we are in a X-point configuration, the plasmas and magnetic fields on both sides of the reconnection point interact locally and only at the X-point site...” Even when reconnection is restricted to a single X-point, it is an overstatement to claim that reconnection is a strictly local phenomenon. One cannot isolate the local magnetic field topology from the plasma geometry in the vicinity of the X-point. Similarly, I cannot agree with the following statement in the rebuttal letter: “In this configuration, the B-fields on the right and left of that X-point do NOT interact. These parts of the plasma feed the contact point where MR takes place, but they do not directly participate to MR in itself.” Magnetic reconnection (MR) is not simply the local change in magnetic topology, but is a plasma process that involves both fields and particles, including flows and gradients. Indeed, the authors appear to be in agreement with this point when they say the following in their rebuttal letter: “Reconnection is a multi-scale phenomenon, that is to say that small scales (ions and electrons) as well as large scales (MHD) play a role in the dynamics of the reconnection event. It is on these larger MHD scales that the geometry is particularly relevant. Please note that the distinction between X-point and elongated MR—as discussed in the rebuttal letter—was never a point of confusion regarding the original presentation. The non-locality of the reconnection process holds for both cases.

We agree with the reviewer that the assessment related to the local reconnection is indeed an oversimplification, since reconnection takes place over an extended zone in the field reversal region. Such a statement has been replaced and in fact the whole discussion is now reshaped (see the modified text in the paper, which is in **purple**) in view of reconnection not being limited to a contact point. We thank the reviewers for having pushed us to be more careful about our analysis and consideration of the problem.

The addition of the face-on view (c) and 3D view (d) in the schematic of Fig. 1 do add some additional context to the geometry of the experiment. However,

the side view (a) is still somewhat misleading. What is shown is not the configuration as viewed side-on from a distance, but rather a slice through the configuration in the x - y plane at $z = 0$. If the slice were made at some other (small) value of z , the orange rectangles (foils T2 and T3) would be offset horizontally from one another.

To better illustrate my concerns, I have prepared a figure (below) that contains what I believe to be a more global depiction of the 3D geometry of the interaction region, based upon the information in panel (d) of Fig. 1 in the manuscript. While the top view is very similar to panel (a) of Fig. 1 in the manuscript, the side view and 3D view show more clearly the geometry of the interaction region when the two foils are rotated relative to one another.

Figure 1: Schematic representation of experimental geometry using 3D surfaces. The red arrows indicate the direction of B .

We thank the reviewer for the efforts she/he made in answering to us, and by providing us with a new point of view of how to depict in a clearer manner the complex geometry of the setup. We have followed the reviewer's suggestion and tried to clarify it now in Figure 1 of the main text, which incorporates a new 3D view, similar to that prepared by the reviewer.

While the rotation of the foils does result in there being an effective guide field at the point of contact between the two outermost rings, it is not self evident that the simultaneous change in local geometry at the contact point will not influence the reconnection dynamics. Thus, ascribing the dependence on angle θ to the magnitude of the guide field alone is an oversimplification. Based on the preceding considerations, I find the following sentence of the Abstract (line 30) to be misleading: “Here we test, in the laboratory and using the canonical “X-point” configuration, where the two magnetized plasmas meet at a single point, the effect of imposing an out-of-plane (guide) magnetic field which breaks the initial symmetry of the system.” It would be more appropriate to say that the guide field arises as a consequence of the geometry of the setup than to say that it is imposed.

We agree that the term “impose” may be confusing in the regard of the typical guide field reconnection. In such a numerical study, the guide field is indeed imposed. The guide field in the experiment is a result of the imposed geometry. Consequently we modified all this wording in the main text, now referring to a guide field naturally arising out of the geometry of the experiment.

I wish to reiterate that I consider the authors to have provided a thorough and accurate analysis of their experimental results. My reservations lie not with the analysis, but with the interpretation—especially with the attribution of differences due to variation of the angle θ to the change in guide field alone. In other words, I remain unconvinced by the authors assertion (from the rebuttal letter) that “In the end, we believe that the demonstration we bring of the surprisingly strong influence of a guide-field on the dynamics of a MR event, unspoiled by other factors, could be of interest to the broad community interested in MR and magnetic field dynamics in plasmas.” [emphasis added.] Because reconnect depends on the characteristics of the plasma beyond the immediate vicinity of the X-point, variation in “other factors” must be taken into account.

We agree with the reviewer and thank her/him for having pushed us to now perform full 3D simulations that allowed us to clarify the respective role of the guide field, which is the dominant effect of the slow down of the magnetic reconnection, and to highlight the expansion of the current sheet along the x-axis (laser axis).

Regarding the role of simulations in this manuscript, I accept the authors’ assertion that “First, we would like to underline that the evidence put forward in the paper regarding the strong slowing down of MR on a X- point magnetic reconnection event is based on experimental data, irrespective of any numerical simulations.” However, it must be kept in mind that a 2D-3V simulation is, in effect, infinite but uniform in the third spatial dimension, which is not a particularly good representation of a plasma confined to a pair

of disks (irrespective of the angle θ). The fact that the 2D simulations show a inhibitory effect of adding a guide field is significant. Indeed, I certainly do not mean to suggest that a guide field cannot have such an effect. It is the degree of the effect that is at question (see next paragraph). Again, my main point is that other contributing factors cannot be ignored when interpreting the experimental results.

Again, we would like to thank the reviewer for stressing that using 2D-3V simulations can present an oversimplified picture and may miss other effects. As detailed by the reviewer, the third direction of the simulation is then treated as infinite. However, in the experiment along the third direction, a strong gradient of density and temperature are present, which may imply a different picture. This point made by the reviewer brought us to run 3D hybrid-PIC simulations, in which the expansion of the laser-driven plasma has been mocked up.

From this new set of 3D simulations, we observed that the reconnection is mediated by a Biermann-battery effect, as already observed by Matteucci et al. (2018), i.e. the gradient of electron temperature and the gradient of density drives the reconnection. The 3D simulations show a similar behavior of the system: when the geometry is tilted, we observe a loss of efficiency caused by the presence of the naturally-arising guide field.

As the authors point out in their introduction, past simulation studies of guide-field reconnection have yielded somewhat contradictory results. Since not all of these studies involve reconnection at an X-point, I will focus on Refs. [16] and [20], both of which do simulate X-point reconnection. While Ref. [16] only considers guide-field ratios $B_g/B_0 = 1$ and 5, Ref. [20] treats guide-fields ratios $B_g/B_0 < 1$ as well. Reference [16] is nevertheless significant because it extends the “GEM Reconnection Challenge” study so as to include a guide field [“Geospace Environmental Modeling (GEM) magnetic reconnection challenge,” J. Birn et al., J. Geo-phys. Res., 106, (2001) doi:10.1029/1999JA900449]. Both studies [16] and [20] find X-point reconnection to be somewhat inhibited by the presence of a guide field, but neither is consistent with the following statement in the Abstract of this manuscript: “A strong slowing down and even halting of reconnection process is observed, even for a weak guide-field.” If the claim that the experimental results presented in this manuscript are valid for X-point reconnection in general, then once must conclude that the aforementioned simulation studies are therefore invalid. Alternatively, once must assume that there is some fundamental difference in the dynamics and geometry of the respective reconnection scenarios that accounts for the differences. I believe the latter interpretation is the correct one. In either case, the final sentence of the abstract, (“These observations stress the importance of taking into account guide-field effects in reconnection taking place in natural and laboratory environments.”) is a point well taken, albeit one that has been long recognized

as well. In this regard, it should be noted that the presence of a guide field plays a more significant role in suppressing asymmetric reconnection [“Diamagnetic suppression of component magnetic reconnection at the magnetopause,” M. Swisdak et al., *J. Geophys. Res.*, 108, (2003), <https://doi.org/10.1029/2002JA009726>], for reasons other than those discussed in this manuscript.

By being more precise on the geometry used in the present study, and on the factors influencing reconnection, we hope that the conversation on this topic can be advanced by our experimental findings, and to allow a reader to compare what was inferred in previous studies with our results.

In my original report, I suggested an alternative title that would more accurately describe the experiment. As the authors have pointed out, the interaction is between disks rather than cylinders. I appreciate this clarification. However, after taking that clarification into account, I still believe that a revised title along the lines of the following would be a more accurate description: “Laboratory evidence of the hampered X-point magnetic reconnection at the interface between non-parallel expanding plasma disks with embedded azimuthal magnetic fields.” To some degree, the fact that that plasma is localized to adjacent disks makes it harder to envision a physical scenario (e.g., in a space or astrophysical context) where such a geometry would provide a natural model. (Adjacent cylinders, on the other hand, suggest scenarios such as interacting flux ropes in the solar corona, for example.) The authors are, of course, welcome to provide illustrative examples of naturally occurring regimes where their experimental geometry would be a good approximation.

Regarding the issue of the applicability of the topology used in the present study with respect to space or astrophysical phenomena, we would like to point out the following: In the solar atmosphere, a large variety of magnetic configurations can develop. Some can also result in an eruptive event such as a solar flare or coronal mass ejection (CME).

From what we can gather from the literature, magnetic reconnection is seen as one of the leading candidates for those eruptive events and even explains the evolution of magnetic topology as the twisting of magnetic flux. A number of effects and topology features can affect the topological evolution and result in a build-up of energy at the magnetic inversion line. Magnetic shearing (or guide field) is one of them. Since the establishment of the CSHKP model (Carmichael, 1964; Sturrock, 1966; Hirayama, 1974; Kopp and Pneuman, 1976), other alternative models have been introduced to explain how the sudden release of energy occurs or what is the cause of the MR onset. Fig. 2 and 3 present only a couple of these models, where

the magnetic lines are tilted one compared to the other one, and we believe that the geometry used in our study precisely relates to those configurations.

Figure 2: 3D illustration of the reversed-shear model. Panels a, b, and c represent the tearing instability phase, the collapsing phase, and the erupting phase, respectively. Reproduced from Birn et al.,(2007), Fig. 5.39, page 280 from book "Reconnection of Magnetic Fields Magnetohydrodynamics and Collisionless Theory and Observations" EDITORS: J. Birn, Los Alamos National Laboratory, E. R. Priest, University of St Andrews, Scotland DATE PUBLISHED: March 2007, ISBN: 9780521854207.

Figure 3: standard model for the magnetic field explosion in single-bipole eruptive solar events, where the bipoles having sigmoidally sheared and twisted core fields and accommodate confined explosions as well as ejective explosions. The rudiments of the field configuration are shown before, during the onset of an explosion that is unleashed by internal tether-cutting reconnection. The dashed curve is the photospheric neutral line, the dividing line between the two opposite-polarity domains of the bipole's magnetic roots. The ragged arc in the background is the chromospheric limb. The gray areas are bright patches or ribbons of flare emission in

the chromosphere at the feet of reconnected field lines, field lines that we would expect to see illuminated in SXT images. The diagonally lined feature above the neutral line in the top left panel is the filament of chromospheric temperature plasma that is often present in sheared core fields. Reproduced from Moore et al., (2001), *THE ASTROPHYSICAL JOURNAL*, 552 : 833-848, 2001 May 10, © 2001. The American Astronomical Society. All rights reserved. Printed in U.S.A. "ONSET OF THE MAGNETIC EXPLOSION IN SOLAR FLARES AND CORONAL MASS EJECTIONS". <https://doi.org/10.1086/320559>

The two investigations mentioned above try to provide overall pictures of the magnetic topology mediated by the reconnection. However, they don't emphasize how the reconnection occurs at a microscopic or mesoscopic scale. For instance, theoretical works [Wright 2020, Priest 2020] try to disentangle how magnetic tubes get twisted while conserving magnetic helicity. At the same time, numerical works emphasize understanding how the interaction of two untwisted magnetic tubes behaves and may result in a twisting of those two magnetic tubes, see, Figure 3. Again, such configuration seems to be closely related to the one we used in our experiment.

Figure 4: Magnetic field isosurfaces at $jB_{jmax}=3$ of reconnecting, untwisted flux tubes in simulation A, at $tvA \ \nu_i / R = [0 ; 1000 ; 1320 ; 1470 ; 1970 ; 3040]$. The tubes are pushed together by a stagnation point flow at $vA=30$ and flatten out on contact (panel [b]) because they have no twist to maintain their cylindrical cross sections. As a result of the magnetic resistivity, at $S \ 1/4 \ 5600$, they then reconnect at several

locations in the three-dimensional equivalent of the tearing mode instability (panels [c] and [d]). Finally, the reconnected portion of the flux merges into a single flux tube (panels [e] and [f]) in the three-dimensional equivalent of the coalescence instability. Reproduced from Linton et al.,(2003), The Astrophysical Journal, 595:1259-1276, 2003 October 1, © 2003. The American Astronomical Society. All rights reserved. Printed in U.S.A. “Three-dimensional Reconnection of Untwisted Magnetic Flux Tubes”. <https://doi.org/10.1086/377439>

In addition to solar physics, a tilted configuration of the magnetic ribbons is found to have an analogy also with the space physics study of Oiesoret et al. [Oiesoret, 2019]. However, the geometry shows discrepancies in the sense of how the magnetic fluxes are entangled. In this latter study, the guide field arises from geometry.

In short, with the examples given above, we believe that indeed our configuration of two tilted magnetic field structures has relation to naturally-occurring events, such as in the solar corona. This is now detailed (see the modified text in the paper in purple) and added to the reshaped introduction of the paper.

In summary, if the experiment described in this manuscript were in fact—as the authors claim—representative of X-point reconnection in general, then this work would certainly be appropriate for publication in Nature Communications. However, as detailed in my report, I believe the claim of general applicability is overstated. Provided that the claims implying the generality of the experimental results are sufficiently softened, I would be happy to recommend publication of this study. Before concluding that Nature Communications is (or is not) the appropriate venue for publication, I wish to afford the authors the opportunity to make the necessary modifications and to then argue their case in this regard.

We thank the reviewer for his support regarding the interest of our findings. We also thank the reviewer for having pushed us in making the effort of confronting our experimental results to full 3D modeling, which helped us grab a better view of the physics at play and of the importance of non-local effects, such as shearing, for large tilts. As detailed in our answers to the reviewer’s comments, we have profoundly reshaped the discussion in the paper in view of the new 3D modelling, and have presented examples (such as those detailed above) of natural/solar configurations that we believe are akin the one used in our experiment. We hope the reviewer finds our results now sufficiently supported and valuable for the community at large.

Reviewer #2 (Remarks to the Author):

Report on the revised NCOMMS-19-26005A, “Laboratory evidence of the halting magnetic reconnection by a weak guide field”, by S. Bolanos et al.

The authors have made attempts to address some concerns that I have raised in my previous report but I do not think they were successful. The point is not about the shape of the diffusion region as shown in Fig.A in their reply; it is about whether the reconnection takes place in 2D or 3D. In 2D, there is an invariant direction along which everything occurs in exactly the same way while in 3D, there does not exist such a direction. I have explained the differences in the points #1 and #2 in my previous report that I will not repeat here. Therefore, reconnection occurs in fundamentally different ways in 2D vs 3D, and the 2D understanding cannot be applied to 3D without justifications. Such justifications should not be easy as I explained in detail in point #2 in my previous report. The experimental setup is highly 3D and reconnection occurs only at one point in the 3D space, with a lot of freedom for plasma to flow around about it. In order to develop a correct physics interpretation of the experimental results, the 3D geometry needs to be taken into account. Therefore, I cannot recommend the acceptance of this manuscript in the present form for publications in Nature Communications.

We agree with the reviewer that the assessment related to reconnection taking place mostly in 2D was indeed an oversimplification, and that 3D effects influence the physics at play here. We also thank the reviewer for having pushed us to be more careful about our analysis and consideration of the problem. As a result, we made the effort of confronting our experimental results to full 3D modeling, which helped us grab a better view of the physics at play and of the importance of 3D. The new 3D simulations in the paper are presented in Fig.1 and 4-6.

Overall, we believe that these new 3D simulations will address and answer the main preoccupation of the reviewer, i.e. that the previously presented 2D simulations could miss an essential element of the physics at play. Essentially, these new 3D simulations concur with the message of the previous 2D simulations, i.e. the effect of the guide field is predominant in reducing the efficiency of reconnection.

In short, we use the same geometry in the 3D simulations than in the 3D experiment. And it results in a similar observation of compressed magnetic field, increased electron density and delayed heating when we have the addition of a guide-field in the setup. This is shown in the new Fig.1 and 4-5.

This analysis is detailed in the paper (see all the modified text in the paper, which is in purple), which has been profoundly reshaped. With this, we hope the reviewer finds our results now sufficiently supported and valuable for the community at large.

Reviewer #3 (Remarks to the Author):

**I would like to thank the authors for their explanations.
I believe the point I raised have been addressed in a reasonable manner.**

We thank the reviewer for his/her support.

In response to the suggestions of reviewers 1 and 2, we made a strong effort to move toward 3D simulations. As discussed in detail in the new version of the text, the new simulations essentially show that the reconnection is mediated by the Biermann-battery, which can only be revealed in a 3D simulation. Nonetheless, the weak GF (as previously revealed by our 2D simulations) is the main effect of restraining the rate at which reconnection unfolds.

**There is one typo that stood out: line 365, the words "loosing efficiency" should probably be "losing efficiency".
But otherwise I personally think the paper is ready to be published.**

This is now corrected, thanks.

While not fully within my area of expertise and quite technical, I believe the draft will be of interest to a fairly wide plasma physics audience.

We thank the reviewer for his/her support.

REVIEWER COMMENTS

Reviewer #1 (Remarks to the Author):

Please see uploaded PDF file.

The latest revisions to Manuscript NCOMMS-19-26005B-Z together with the authors' reply to previous comments contain a number of helpful clarifications. For example, the first sentence of the Conclusion section emphasizes that it is the "tilt between two magnetic tori" that controls the behavior of the system. The new panel (d) in Fig. 1 also helps in this regard. Also of particular value is the emphasis in the authors' latest rebuttal letter on the relevance of the experimental configuration to the interpretation of interacting flux ropes in the solar corona. And while the analogy isn't perfect (coronal flux ropes typically have poloidal as well as toroidal magnetic fields whereas in the experiment the fields are only toroidal), this application nevertheless provides a well-motivated physical context. The presentation would therefore be significantly improved if the discussion (from the rebuttal letter) of the connection between the experiment and the geometry of interacting coronal flux ropes was incorporated into the manuscript to a greater degree.

While the authors' experiment may be relevant to interpreting the interaction of obliquely oriented magnetic flux tubes in the solar corona, the relevance to reconnection in the magnetosphere is still not evident based on the references the authors choose to cite. Near the start of the Conclusion section it is stated that "We note that the tilted configuration we explored here is relevant to MR at Earth's magnetopause^{54,55}. . ." However, as mentioned in a previous review, these cited studies are considering the suppression of reconnection due to diamagnetic drift resulting from the interaction of a guide field with a change of plasma B across the current sheet. In other words, this suppression mechanism applies *only* in the case of *asymmetric* reconnection where ΔB as defined in Ref. 55 does not vanish. In the experiment discussed in this manuscript, the two tilted tori are equivalent (i.e., the reconnection is symmetric to the extent that $\Delta B = 0$). Therefore, the cited references do not support the authors' argument as claimed.

As a counterexample of guide-field suppression in the case of *symmetric* reconnection in the magnetosphere, the authors are referred to the following publication by the same first author as Ref. 54: [Øieroset, M., et al. (2016), "MMS observations of large guide field symmetric reconnection between colliding reconnection jets at the center of a magnetic flux rope at the magnetopause, *Geophys. Res. Lett.*, **43**, 5536–5544, doi:10.1002/2016GL069166]. This study complements the authors' statement at the bottom of p. 1 that "conflicting results" regarding the role of a guide field in reconnection have been reported.

In light of such "conflicting results," it seems reasonable to assume that the presence vs absence as well as the strength of the guide field does not *alone* control the ability of magnetic fields to reconnect. As was repeatedly emphasized in previous reviews, the *geometry* of the reconnection region must play a role as well. Therefore, focusing on physical scenarios with similar geometry—such as interacting coronal loops—would provide a narrower but better justified focus for this study. To this end, a title such as "Laboratory evidence of magnetic reconnection hampered in obliquely interacting flux tubes" for example would clarify the domain being modeled while avoiding the implication that the guide field is the *only* relevant factor. The Abstract should also be rewritten accordingly.

When revising the manuscript, the authors should also consider the following issues:

The opening sentence of the Introduction refers to "... many space and astrophysical events." Based on the word "many," the reader might expect more than one reference from three different categories. Furthermore, Ref. 1 would be more accurately described as *solar* since "space" generally implies accessible by in situ probes. Also, Ref. 2 is not astrophysical; and focuses on substorms in general more than on reconnection specifically. I therefore ask the authors to revise this sentence with these points in mind.

There is also some confusing wording in the new (red) text sections:

At the top of p. 2, the meaning of "... which is confronted to three-dimensional hybrid simulations. . ." is unclear.

Near the top of the Results section, "... at the midway of both the plasma. . ." would be clearer as "... midway between the two plasmas. . ." (if that is what is meant).

At the top of p. 6, "These two operators allow to create and sustain an axial electron density gradient. . ." would be clearer as "These two operators allow an axial electron density gradient to be created and sustained. . ."

I am also having difficulty interpreting the black surface in Fig. 1e (described as the "configuration of the targets"). Please include additional clarification.

In summary, I recognized the importance of laboratory experiments as a tool for studying magnetic reconnection. The authors' experiment uses a novel approach and is a valuable addition to the field. At the same time, it does *not* represent a generalized test of the influence of guide fields in reconnection and therefore any implication that it does should be avoided. By emphasizing the similarity of the experimental geometry with that of abutting flux tubes in the solar corona, the relevance of the experiment to a specific but important plasma environment would provide a valuable focus thereby avoiding an implied generalization to domains where the experimental results may not apply.

Reviewer #2 (Remarks to the Author):

Report on the 2nd revised NCOMMS-19-26005A, "Laboratory evidence of magnetic reconnection hampered by a weak guide field", by S. Bolanos et al.

The authors have made a substantial revision of this manuscript by replacing 2D simulation results with their 3D counterpart. The reported simulation results make a lot of sense themselves and also when used in interpreting experimental results. I appreciated authors' effort in making this happen.

As a result, now it is clear that the "mouse"-shaped proton images in Fig.1 when the tilt angle is increased are due to larger magnetic field gradient, shown in Fig.4c. (However, I do see there are still quantitative differences between experimental data and simulation prediction in Fig.1; experimental data show much larger "mouse". Perhaps, such differences can be accounted for by adjusting some simulation parameters.)

However, I do have serious concerns on the primary conclusion of this manuscript, i.e., reconnection is slowed by the guide field. I don't think this is a correct interpretation. Below let me explain why.

Based on the 3D setup reported here, a major difference between anti-parallel case ($\theta=0$) and tilted cases ($\theta>0$) becomes clear: in the anti-parallel case the plasma on T2 and the plasma on T3 are colliding with each other at all target locations along the z axis. In the tilted cases, however, two plasmas collide only over a fraction of the total target distance in the z axis around $z=0$. Larger the tilt angle, smaller fraction of the distance over which two plasmas collide. (I have made this clear in my second point of my first report of this manuscript.)

Now, for the tilted cases when $\theta>0$, it is easy to understand that two plasmas can push closer to each other only around $z=0$ to form larger magnetic gradient, compared with the anti-parallel case where two plasmas collide all along the targets to pile up preventing the formation of larger magnetic gradient. This explains the results shown in Fig.4 and Fig.6. As for Fig.5, it is obvious that larger the tilt angle, smaller fraction of the plasmas is colliding resulting in less dissipation and electron heating. (The other part of plasmas will miss each other outside the fraction around $z=0$.)

From the above analysis, it becomes clear that guide field is not necessarily the reason for less dissipation and electron heating due to slower reconnection, rather it is more likely due to the change in the length over which two plasmas collide, leading to the less overall interactions between two plasmas except the location around $z=0$ when the tilt angle is increased.

Therefore, unfortunately I still cannot recommend the acceptance of this manuscript in the present form for publications in Nature Communications.

Answer to the comments of Reviewer #1

The latest revisions to Manuscript NCOMMS-19-26005B-Z together with the authors' reply to previous comments contain a number of helpful clarifications. For example, the first sentence of the Conclusion section emphasizes that it is the "tilt between two magnetic tori" that controls the behavior of the system. The new panel (d) in Fig. 1 also helps in this regard. Also of particular value is the emphasis in the authors' latest rebuttal letter on the relevance of the experimental configuration to the interpretation of interacting flux ropes in the solar corona. And while the analogy isn't perfect (coronal flux ropes typically have poloidal as well as toroidal magnetic fields whereas in the experiment the fields are only toroidal), this application nevertheless provides a well-motivated physical context. The presentation would therefore be significantly improved if the discussion (from the rebuttal letter) of the connection between the experiment and the geometry of interacting coronal flux ropes was incorporated into the manuscript to a greater degree.

We thank the reviewer for his insight, and for supporting the idea that the overall investigation may interest a larger community. Following his/her suggestion, we have now completed the discussion on the solar context and flux reconnection by adding the following paragraph (the modified text is in purple):

"A factor complicating the picture is the topology of the magnetic fields. Deviating from the idealized picture of the first theoretical models²⁰, in which investigations were restricted to two-dimensional (e.g. in the yz plane) anti-parallel magnetic fields, the magnetic fields in natural three-dimensional events^{21,22} can be decomposed into anti-parallel and possibly an additional component (e.g. along the x-axis) of the magnetic field (commonly called a guide field). A well-known example is that of reconnection in solar arches²³. It involves the emergence of magnetic flux tubes from the convective zone at the solar surface²⁴. These flux tubes can meet and then reconnect. The simplest reconnection picture involves the encounter of two straight flux tubes exhibiting anti-parallel magnetic field. In reality, the situation is rather three-dimensional, either due to the oblique orientation of one flux tube with respect to the other (the configuration explored with our experimental set-up, see below) or due to the twist of the flux tubes along their axis, resulting in the growth of a poloidal component of the magnetic field²⁵ (associated with a current flowing along the core field). The pending challenge the community faces here is to understand how does reconnection take place between two obliquely oriented flux tubes, i.e. in a guide-field reconnection configuration.

Likewise, when leaving the solar corona and cruising toward the Earth, magnetic flux ropes (that is twisted flux tubes) can interact with the Earth magnetopause and also eventually reconnect during Flux Transfer Events (FTE²⁶). As the signature of the

bi-polar component of the magnetic field, normal to the magnetopause, can be clearly identified, magnetic reconnection between two obliquely oriented flux tubes has been put in evidence both by in-situ observations^{8, 27, 28} and numerical simulations^{29, 30}. The modified efficiency of the reconnection process between two obliquely oriented flux tubes then also appears to be a key point in explaining how mass, momentum and energy can be loaded in the inner Earth magnetosphere (or in other planets⁵ like Mars) from the solar wind flowing around.”

While the authors’ experiment may be relevant to interpreting the interaction of obliquely oriented magnetic flux tubes in the solar corona, the relevance to reconnection in the magnetosphere is still not evident based on the references the authors choose to cite. Near the start of the Conclusion section it is stated that “We note that the tilted configuration we explored here is relevant to MR at Earth’s magnetopause[54,55] . . . ” However, as mentioned in a previous review, these cited studies are considering the suppression of reconnection due to diamagnetic drift resulting from the interaction of a guide field with a change of plasma β across the current sheet. In other words, this suppression mechanism applies only in the case of asymmetric reconnection where $\Delta\beta$ as defined in Ref. 55 does not vanish. In the experiment discussed in this manuscript, the two tilted tori are equivalent (i.e., the reconnection is symmetric to the extent that $\Delta\beta = 0$). Therefore, the cited references do not support the authors’ argument as claimed.

As a counterexample of guide-field suppression in the case of symmetric reconnection in the magnetosphere, the authors are referred to the following publication by the same first author as Ref. 54: [Øieroset, M., et al. (2016), “MMS observations of large guide field symmetric reconnection between colliding reconnection jets at the center of a magnetic flux rope at the magnetopause, Geophys. Res. Lett., 43, 5536–5544, doi:10.1002/2016GL069166]. This study complements the authors statement at the bottom of p. 1 that “conflicting results” regarding the role of a guide field in reconnection have been reported.

We agree with the reviewer that indeed the references [54, 55] were misleading and weren’t adequate for the assessment we made. These have now been removed and, instead, we now add the references suggested by the reviewer. We furthermore reworded the paragraph discussing the reconnection process at the magnetopause ; we now include references to flux ropes and flux transfer events, and we also tried to make clear how our experiment on two obliquely oriented flux tubes can compare to the physics at play at the magnetopause.

In light of such “conflicting results,” it seems reasonable to assume that the presence vs absence as well as the strength of the guide field does not alone

control the ability of magnetic fields to reconnect. As was repeatedly emphasized in previous reviews, the geometry of the reconnection region must play a role as well. Therefore, focusing on physical scenarios with similar geometry—such as interacting coronal loops—would provide a narrower but better justified focus for this study. To this end, a title such as “Laboratory evidence of magnetic reconnection hampered in obliquely interacting flux tubes” for example would clarify the domain being modeled while avoiding the implication that the guide field is the only relevant factor. The Abstract should also be rewritten accordingly.

We thank the reviewer for this proposition of title, which we embrace. We believe that this new title provides a better global picture since we recently pointed out a small effect on the 3D geometry that was raised by the other reviewer. For this reason, we also modify as follows the abstract:

*“Magnetic reconnection can occur when two plasmas, having anti-parallel components of the magnetic field, encounter each other. In the reconnection plane, the anti-parallel component of the field is annihilated and its energy released in the plasma. Here, we investigate through laboratory experiments **the reconnection between two flux tubes that are not strictly anti-parallel**. Compression of the **anti-parallel component of the magnetic field** is observed, as well as a decrease of the reconnection efficiency. Concomitantly, we observe delayed plasma heating and enhanced particle acceleration. Three-dimensional hybrid simulations support these observations and highlight the plasma heating inhibition and reconnection efficiency reduction **for these obliquely oriented flux tubes**”.*

When revising the manuscript, the authors should also consider the following issues:

The opening sentence of the Introduction refers to “. . . many space and astrophysical events.” Based on the word “many,” the reader might expect more than one reference from three different categories. Furthermore, Ref. 1 would be more accurately described as solar since “space” generally implies accessible by in situ probes. Also, Ref. 2 is not astrophysical; and focuses on substorms in general more than on reconnection specifically. I therefore ask the authors to revise this sentence with these points in mind.

We thank the reviewer for raising these points. The sentence has been changed:

*“Magnetic reconnection is the subject of intense investigations due to its suspected role in the sudden plasma heating and particle energization observed **in varied spatial and astrophysical events, e.g. solar flares¹⁻³, planetary magnetic substorms⁴⁻⁶, flux transfert events^{7,8}, or black hole plasma jets⁹**.”*

There is also some confusing wording in the new (red) text sections:

- **At the top of p. 2, the meaning of “...which is confronted to three-dimensional hybrid simulations...” is unclear.**

The sentence had been modified to:

“To help progress on those issues, here we highlight, using laboratory measurements and three-dimensional hybrid simulations, ...”

- **Near the top of the Results section, “. . . at the midway of both the plasmas. . . ” would be clearer as “. . . midway between the two plasmas. . . ” (if that is what is meant).**

We agree with the reviewer, and the suggestion has been included.

- **At the top of p. 6, “These two operators allow to create and sustain an axial electron density gradient. . . ” would be clearer as “These two operators allow an axial electron density gradient to be created and sustained. . . ”**

This suggestion has been considered and the sentence has been modified accordingly.

- **I am also having difficulty interpreting the black surface in Fig. 1e (described as the “configuration of the targets”). Please include additional clarification.**

The black surface aims at representing the (initial) solid target, i.e. before the laser irradiation. For the sake of clarity, Figure 1e has been modified, and the following sentence is included in the caption to provide a clear guidance to the reader:

“Snapshot of a simulation in the tilted configuration using the 3D hybrid code AKA: the targets are depicted in gray, and electron density in color, and the black lines represent the magnetic lines.”

In summary, I recognized the importance of laboratory experiments as a tool for studying magnetic reconnection. The authors’ experiment uses a novel approach and is a valuable addition to the field. At the same time, it does not represent a generalized test of the influence of guide fields in reconnection and therefore any implication that it does should be avoided. By emphasizing the similarity of the experimental geometry with that of abutting flux tubes in the solar corona, the relevance of the experiment to a specific but important plasma environment would provide a valuable focus thereby avoiding an implied generalization to domains where the experimental results may not apply.

We thank the reviewer for his/her positive and supportive comments and hope that the modifications brought to the paper make it now suitable for publication.

Answer to the comments of Reviewer #2

The authors have made a substantial revision of this manuscript by replacing 2D simulation results with their 3D counterpart. The reported simulation results make a lot of sense themselves and also when used in interpreting experimental results. I appreciated authors' effort in making this happen.

As a result, now it is clear that the “mouse”-shaped proton images in Fig.1 when the tilt angle is increased are due to larger magnetic field gradient, shown in Fig4c. (However, I do see there are still quantitative differences between experimental data and simulation prediction in Fig.1; experimental data show much larger “mouse”. Perhaps, such differences can be accounted for by adjusting some simulation parameters.)

We want to stress to the reviewer that we do not aim, in order not to overstate the quantitative comparison between simulations and experiment, at having one-to-one correspondence between the synthetic images and the experimental ones. This is why for example the time is given in units of ω^{-1} for the simulation. This said, we now better adjusted the size of the synthetic images, using the coplanar case as a gauge (i.e. matching the length of the black thin line between the experimental and synthetic images). We hope that the reviewer finds the comparison more satisfactory now.

However, I do have serious concerns on the primary conclusion of this manuscript, i.e., reconnection is slowed by the guide field. I don't think this is a correct interpretation. Below let me explain why.

Based on the 3D setup reported here, a major difference between anti-parallel case ($\theta=0$) and tilted cases ($\theta > 0$) becomes clear: in the anti-parallel case the plasma on T2 and the plasma on T3 are colliding with each other at all target locations along the z axis. In the tilted cases, however, two plasmas collide only over a fraction of the total target distance in the z axis around $z=0$. Larger the tilt angle, smaller fraction of the distance over which two plasmas collide. (I have made this clear in my second point of my first report of this manuscript.)

Now, for the tilted cases when $\theta > 0$, it is easy to understand that two plasmas can push closer to each other only around $z=0$ to form larger magnetic gradient, compared with the anti-parallel case where two plasmas collide all along the targets to pile up preventing the formation of larger magnetic gradient. This explains the results shown in Fig.4 and Fig.6. As for Fig.5, it is obvious that larger the tilt angle, smaller fraction of the plasmas is colliding resulting in less dissipation and electron heating. (The other part of plasmas will miss each other outside the fraction around $z=0$.)

From the above analysis, it becomes clear that guide field is not necessarily the reason for less dissipation and electron heating due to slower reconnection, rather it is more likely due to the change in the length over which two plasmas collide, leading to the less overall interactions between two plasmas except the location around $z=0$ when the tilt angle is increased.

Therefore, unfortunately I still cannot recommend the acceptance of this manuscript in the present form for publications in Nature Communications.

We thank the reviewer for highlighting this crucial issue in the analysis of the results, and giving us the opportunity to quantitatively investigate it and clarify the whole issue of the structure of the magnetic field. The reviewer will find below three figures that should convince her/him that the volume of the interaction between the two plasmas, as well as the length of the current sheet is very comparable in the non-tilted and tilted cases, thus supporting our interpretation of the reduced efficiency of the reconnection (when tilting the targets) as mainly due to the effect of the arising guide field.

Figure A (below) displays the current density J_x in the xy plane, at $z=0$ and $z=15d_0$. It clearly appears that these lengths are very comparable, whatever the z value (for a given theta angle). The reason for this is that the plasma plumes that are created span over several tens of inertial length along the X -direction.

We want to make it clear, as this is the crux here, that the similar length of the current sheet when tilting the targets is due to the fact that the magnetic field produced by the Biermann-battery mechanism is frozen in the plasma plume, *especially at the outer edge of the plume* (see the detailed characterization on this specific point provided by Campbell et al., PRL 2020, ref. 37 of the paper). Consequently, the B-field will expand in a similar manner as the plasma plume (having a somewhat hemispherical shape), over *several tens of ion inertial lengths, i.e. constitute a rather "thick" magnetic structure.*

Fig. A: Out-of-plane current J_x , at $t=30_0^{-1}$, in two planes of the 3D simulation; 1st row shows the xy plane at $z=0$, 2nd row is at $z=15d_0$.

To demonstrate this, and in order to go beyond the visual impression provided by Fig.A, we have quantitatively computed the volume of the diffusion region by calculating the volume of the current sheet at the reconnection site. Fig. B shows the results of such a calculation. It highlights that the diffusion region volume is very similar for all the configurations. It demonstrates that for a small angle, the geometry effect is still negligible but going to an angle higher than 45° , the geometrical effect has to be included. The panel (b) of Fig.B, which is our new analysis, is now included in the paper as Fig.5.c.

Fig. B: (a) Simulated power of the energy conversion from magnetic energy into plasma energy in the electron frame. (b) Evolution of the volume with the integrand from (a) being greater than $\epsilon=1e-8$ (i.e. the numerical noise level). The integrated volume in both cases is limited in the y direction over $-5 < y/d0 < 5$.

Further, this evidence provided by the numerical simulations meets the one of the experimental observations. Indeed, as shown below in Fig.C, the experimental interaction length (here the length of the heated region, following reconnection) diagnosed by the optical pyrometry is similar when tilting or not the targets (see Fig.2 of the paper and Fig.C below). Moreover, this is also witnessed by the proton radiography diagnostic: the length of the mouth or line along the current sheet is not significantly varying when tilting the targets. The averaged length on the proton radiography images is 0.122 mm, 0.117 mm, 0.125 mm and 0.11 mm for respectively 0°, 15°, 30°, and 45° tilt angle.

Fig.C: Length of the heated plasma region (following reconnection), as deduced from self-emission measurements (see Fig.2 of the main paper). The plot represents the time-averaged length of the heated region (the counts are normalized by their respective maximum value).

All this is summarized and emphasized in the paper, when describing the plasma structure in the “discussion” section. We have now added a new paragraph to address the reviewer's concern in detail. When discussing this point, we also now make reference to Fig.5c and Fig 6 where it also clearly appears that the Z-extension of the current sheet does not depend on the tilt angle.

It is as follows:

“We checked that the observed reduced reconnection efficiency is not merely due to a geometrical effect, i.e. we checked that the interaction volume between the two plasma plumes is not sensibly modified as the two targets are tilted. In order to verify this, we computed the temporal evolution of the diffusion volume, see Fig.5.c. It shows that the interaction volume is weakly affected when tilting the targets. Similarly, Fig. 6 clearly shows that the current sheet length is not depending on the θ angle. The numerical observations are further reinforced by the experimental observations that (i) the length of the magnetic interaction zone between the two flux tubes as well as (ii) that of the heated plasma (following reconnection) vary both weakly as a function of the tilt angle between the two targets, see Fig.1(e-g) and Fig.2(a-c) respectively. All this is due to the fact that, while there is an axial gradient (i.e. along the x -direction) of the plasma parameters, the associated B_z magnetic field is weakly depending on the x location. It is so because the height of the plasma plume, at its edge⁴⁰, is larger than its radius, so each flux tube is facing its twins with locally comparable values of the plasma parameters. As a consequence, and as

evidenced in both the experiment and the simulations, even in the tilted cases, the length (along the z axis) of the current sheet is weakly depending on the x coordinate, whatever the θ angle.”

With these modifications brought to the paper, we hope that the reviewer will find it now suitable for publication.

REVIEWER COMMENTS

Reviewer #1 (Remarks to the Author):

Please see uploaded PDF file.

The latest version of Manuscript NCOMMS-19-26005C contains a number of significant modifications that greatly reduce the possibility of misinterpreting the experimental results. A few minor request for further clarification will be addressed later.

Among the revisions, I find the new panel (d) of Fig. 1 to be particularly informative. This visualization of the interacting plasma plumes from the 3D simulation used to model the experiment conveys more about the geometry of the interaction than can easily be described with words or 2D plots alone. In fact, I think the manuscript would benefit if more such figures were included. In particular it would be useful to see this volume rendering from a viewpoint approximately parallel to the x axis looking back toward the targets— with a corresponding plot for the $\vartheta = 0$ case. It is clear from Fig. 1(d) that the plumes originating from the two tilted foils expand roughly along the directions *normal* to the plane of each respective foil. This geometry suggest an inhomogeneity introduced by the tilt that can potentially influence the reconnection rate.

The schematic in Fig 1 below illustrates this geometric effect, where units are normalized to the distance between the centers of the two heated regions (yellow and cyan) at $y = \pm 0.5$ and $x = z = 0$. The green and blue line segments are normal to the two foils at the center of each heated zone marking the axes of the expanding plumes.

Figure 1:

The red dotted line segment connects the two normals where they cross the $x = 1$ plane. The length of this segment (for arbitrary x) is $[1 + 4x^2 \tan^2(\vartheta/2)]^{1/2}$, and so increases with x except for the case $\vartheta = 0$. It would seem reasonable to assume that the contribution to the reconnection rate from any given plane of constant x would depend to some degree on this separation. This x -dependent inhomogeneity is, at a minimum, a potential contributing factor to the tilt-angle dependence of reconnection in both the experiment and the 3D simulation—irrespective of the argument put forward regarding the relative insensitivity of the “interaction volume” on the tilt angle.

In relation to the previous point, is stated that in the simulation, “The distance between the focal spot centers is $56d_0$.” Therefore, the position of the cuts in Fig. 6 of $10d_0$ is only $\sim 18\%$ of the inter-spot

separation. To better understand how the various quantities plotted depend on distance from the source of the plasma plumes, it would be informative to supplement Fig. 6 with cuts for at least one additional value of x significantly further from the targets.

The inhomogeneity introduced by the tilt presumably relates to the inhomogeneity referred to in the following text from near the end of the introduction: “Note also that the guide field is here inhomogeneous, which is different than the setup of most numerical studies [Refs]. There is however no clear reason why the out-of-plane magnetic field would be uniform in natural events and not more alike our experimental setup. . .” It is certainly reasonable to assume that *some* natural events exhibit an inhomogeneous guide field as in the present experimental setup. However, this doesn’t preclude the existence of other natural events that are better represented by the *homogeneous* guide field of the past numerical studies referenced in the quoted text. As long as it is clearly stated that the findings are tied to a specific experimental scenario, there should be no ambiguity. However, one does need to consider the role inhomogeneity itself might play in affecting the reconnection rate. In other words, it cannot be assumed that a homogeneous guide field and an inhomogeneous guide field affect the reconnection rate in the same way.

While the revised text is much improved, there is still room for additional clarification. For example, the second paragraph of the **Introduction** begins as follows: “A factor complicating the picture is the topology of the magnetic fields. Deviating from the idealized picture of the first theoretical models [Refs], in which investigations were restricted to two-dimensional (e.g. in the yz plane) anti-parallel magnetic fields, the magnetic fields in natural three-dimensional events [Refs] can be decomposed into anti-parallel and possibly an additional component (e.g. along the x -axis) of the magnetic field (commonly called a guide field).” As written, this description could be interpreted as implying that a 3D geometry is *necessary* for the inclusion of a guide field. I’m sure the authors did not mean to suggest this to be the case since the orientation of \mathbf{B} in a 2D simulation is not restricted to lie in the simulation plane. Please reword these sentences to avoid any ambiguity on this point.

The same paragraph ends as follows: “The pending challenge the community faces here is to understand how does reconnection take place between two obliquely oriented flux tubes, i.e. in a guide-field reconnection configuration.” This sentence suggests that the “guide-field reconnection configuration” is the *defining* property of obliquely oriented flux tubes rather than one of many characteristic properties. This distinction could be made more clearly by replacing “i.e.” with a more flexible transition (for example, “. . . flux tubes, which includes the presence of a guide field.”).

In summary, this manuscript now clearly presents the key experimental results and their relevance to a particular domain of naturally occurring plasma interactions without overstating the generality of the finding to all scenarios in which guide-field reconnection can occur. As pointed out, there remain a few cases where minor revisions to the wording would help to further avoid potential misunderstanding or ambiguity. As previously mentioned, I also believe that a few additional figures (3D volume rendered views of the simulation results similar to Fig. 1d as well as additional panels in Fig. 6 for a larger value or values of x) would aid the reader in visualizing the inhomogeneous nature of the interacting plasma plumes. Pending these modifications, I believe this work should meet the criteria for publication in *Nature Communications*.

Reviewer #2 (Remarks to the Author):

The authors have made adequate improvements to the manuscript so the issues that I have raised previously have been more or less resolved. The key information missing from the previous versions is that the plumes made by laser hitting the target have a substantial extent in the x direction (the target normal direction), close to hemispheres. In the 2nd version of this manuscript, Fig.1(d) 3D view of the thin disk shaped flux tubes gave incorrect information about the shape whereas in this new version, Fig.1(d) simulation snapshot provides 3D shapes of electron density and magnetic field lines, with substantial extents in the x-direction. Quantitative analysis added confirmed this picture, at least within the angle range that is reported here. The improved synthetic images of the “mouse” feature were provided. Therefore, now I recommend the acceptance for publication.

Answers to the comments made by Reviewer 1

The latest version of Manuscript NCOMMS-19-26005C contains a number of significant modifications that greatly reduce the possibility of misinterpreting the experimental results. A few minor request for further clarification will be addressed later.

Among the revisions, I find the new panel (d) of Fig. 1 to be particularly informative. This visualization of the interacting plasma plumes from the 3D simulation used to model the experiment conveys more about the geometry of the interaction than can easily be described with words or 2D plots alone. In fact, I think the manuscript would benefit if more such figures were included. In particular it would be useful to see this volume rendering from a viewpoint approximately parallel to the x axis looking back toward the targets— with a corresponding plot for the $\theta = 0$ case. It is clear from Fig. 1(d) that the plumes originating from the two tilted foils expand roughly along the directions normal to the plane of each respective foil. This geometry suggest an inhomogeneity introduced by the tilt that can potentially influence the reconnection rate.

We thank the reviewer for his/her comments. We added additional 3D rendering views of the simulations in Fig. 1, following the suggestion of the reviewer, to provide a clearer understanding of the geometry. We hope that these bring the needed clarifications.

The schematic in Fig 1 below illustrates this geometric effect, where units are normalized to the distance between the centers of the two heated regions (yellow and cyan) at $y = \pm 0.5$ and $x = z = 0$. The green and blue line segments are normal to the two foils at the center of each heated zone marking the axes of the expanding plumes.

The red dotted line segment connects the two normals where they cross the $x = 1$ plane. The length of this segment (for arbitrary x) is $[1+4x^2 \tan^2(\theta/2)]^{1/2}$, and so increases with x except for the case $\theta = 0$. It would seem reasonable to assume that the contribution to the reconnection rate from any given plane of constant x would depend to some degree on this separation. This x -dependent inhomogeneity is, at a minimum, a potential contributing factor to the tilt-angle dependence of reconnection in both the experiment and the 3D simulation—irrespective of the argument put forward regarding the relative insensitivity of the “interaction volume” on the tilt angle.

In relation to the previous point, it is stated that in the simulation, “The distance between the focal spot centers is $56d_0$.” Therefore, the position of the cuts in Fig. 6 of $10d_0$ is only $\sim 18\%$ of the inter-spot separation. To better understand how the various quantities plotted depend on distance from the source of the plasma plumes, it would be informative to supplement Fig. 6 with cuts for at least one additional value of x significantly further from the targets.

The reviewer is right to have us clarify the question of the influence of the geometrical factors.

- 1) Let's first discuss the quantitative changes induced by the geometry. The red segment introduced by the reviewer can be expressed as: $D = [D_0^2 + 4x^2 \tan^2(\theta/2)]^{1/2}$, where D_0 is the distance between the two spots at the surface of the target, i.e. $56 d_0$. At $x = 10 d_0$ and $\theta = 45^\circ$, we find $D = 56.6 d_0$, which corresponds to a 1% increase compared to D_0 . At $x = 25 d_0$, which is just below the tip of the plasma plume, this separation corresponds to a $\sim 7\%$ increase of the in-plane separation ($56 d_0$). Thus, the increase of the effective separation between the plumes is just a geometrical effect and is modest in all cases, and its magnitude does not seem to be able to explain the

clear and quantitative changes we measure in the experimental and simulated observables when tilting the targets. In conclusion, it seems to us that such a small increase of separation of few % (in the tilted configuration) cannot be the main reason behind the observed destabilization of the current sheet.

We make this now clear in the paper (on p.2) as follows:

“Note that the effective distance between the two plasma plumes will increase when tilting the two targets. However, in our conditions, this distance is only increased by less than 7% at maximum (for the 45° tilt) compared to the separation between the two laser impacts in the coplanar configuration. Such modest increase is much lower than any quantitative change observed in the experimental and simulated observables.”

- 2) Second, we have computed the change of the reconnection efficiency along the x axis. This is done by computing the integral of $j(E + v_e \times B)$ in the yz plane, at every x, and this at $t = 30 \Omega_0^{-1}$. The result is shown in Figure A below. We can observe that the efficiency for the coplanar and small angle tilt (15° and 30°) have a similar trend, i.e. it rises as a function of x more or less in the same manner. Note that this is true although the overall efficiency at 30° is reduced with respect to the coplanar case, consistently the evaluation of the overall efficiency (integrated over the whole volume) that is shown in Fig 5a of the main paper.

Now, for the 45°-tilt configuration, not only is the overall efficiency even more reduced, but the point is that reconnection becomes somewhat efficient only at larger x values. In that case (45° tilt), we can thus say that, close to the target ($x < 9 d_0$), the significant drop in efficiency can indeed be attributed to the geometrical aspect. This is also what can be seen in Fig.5c, i.e. the reduction of the volume where reconnection takes place. We see here, with Fig.B that this reduction is mostly a reduction of the zone at low x values where reconnection can take place. To explain this, we note that the surface of the target constrains the length of the current sheet close to the surface. Such a geometrical effect is effective for $x < L_{CS} / (2 \cdot \tan(90 - \theta/2))$, where the L_{CS} is the current sheet length. This effect is negligible for the small angle tilt (15° and 30°), but it has an effect for $x < 9 d_0$ in the 45° configuration as shown in Fig. A (with $L_{CS} \sim 40 d_0$).

Fig. A: x -dependency of the reconnection efficiency: dissipation term ($j \cdot (E + v_e B)$) integrated over the yz -plane at $t = 30 \Omega_0^{-1}$.

We now include Fig.A as a new panel in Fig. 5 of the main paper. The following sentences have been added in the manuscript to describe it.

“Similarly, Fig. 6 clearly shows that the current sheet length is not depending on the angle **except close to the target, where the surface of the target will constrain the length of the current sheet (not shown here). This geometrical aspect is effective for $x < L_{CS} / (2 \cdot \tan(90 - \theta/2))$, where the L_{CS} is the current sheet length. It is negligible for the small angle tilt configurations and has an effect for $x < 9 d_0$ in the 45° configuration, see Fig.5.(d). Fig.5.(d) shows the x -dependency of the reconnection efficiency. In a similar manner to Fig.5.(a), we integrated the dissipation term over the yz -plane (as opposed to Fig. 5(a), where it is volume-integrated) to assess how the reconnection is dependent on x . The efficiency for the coplanar and small angle tilt (15° and 30°) cases have a similar trend, i.e. it rises as a function of x more or less in the same manner. Note that this is true although the overall efficiency at 30° is reduced with respect to the coplanar case. For the 45° -tilt configuration, we notice that not only is the overall efficiency even more reduced, but it clearly becomes efficient only at larger x values than for the other cases. This loss of efficiency is significant close to the target ($x < 9d_0$); we attribute it to the geometrical aspect discussed above, i.e. the fact that the target tilt reduces, close to the surface of the target, the length of interaction of both plasmas.”**

3) Finally, we've now added to the paper (these are the new panels (g-i) of Fig.6) further 2D cut of the current density, just below the tip of the plasma plume (at $x = 25 d_0$). Note that, as we move further away from the target, the flow velocity changes its orientation to be more aligned to the normal of the target, due to the expansion of the plasma plume. In turn, the advection of the magnetic field in the reconnection region is reduced, meaning that the magnetic compression is less effective close to the top of the plasma plume. With a reduced magnetic pressure, the current sheet is less stable to instabilities as shear-flow instabilities in the tilted configuration, which is what is observed. This is Fig.B, which is shown below.

Fig. B: 2D cuts in the yz -plane) of the out-of-plane current J_x at $x = 10 d_0$ for panels (d-f) and at $x = 20 d_0$ for panels (d-f). All the images are retrieved from the 3D simulations, at $t\Omega_0 = 30$.}

The following sentences have been added in the manuscript to describe the new 2D cut of the current density:

“Although we do not observe any difference in the length of the current sheet for the various configurations, however, as one moves farther away from the target surface, the current sheet becomes more unstable in the tilted configuration, see Fig.6(g-i). This is caused by the fact that close to the top of the plasma plumes, the plasma advection is mainly oriented normal to the target. This leads to less magnetic flux being advected toward the current sheet, thus reducing the magnetic compression and magnetic pressure, as seen in Fig. 4. With a reduced magnetic pressure, the current sheet is more prone to other processes such as shear-flow instability [64].

The inhomogeneity introduced by the tilt presumably relates to the inhomogeneity referred to in the following text from near the end of the introduction: “Note also that

the guide field is here inhomogeneous, which is different than the setup of most numerical studies [Refs]. There is however no clear reason why the out-of- plane magnetic field would be uniform in natural events and not more alike our experimental setup. . . ” It is certainly reasonable to assume that some natural events exhibit an inhomogeneous guide field as in the present experimental setup. However, this doesn’t preclude the existence of other natural events that are better represented by the homogeneous guide field of the past numerical studies referenced in the quoted text. As long as it is clearly stated that the findings are tied to a specific experimental scenario, there should be no ambiguity. However, one does need to consider the role inhomogeneity itself might play in affecting the reconnection rate. In other words, it cannot be assumed that a homogeneous guide field and an inhomogeneous guide field affect the reconnection rate in the same way.

We agree with the statement of the Reviewer and that the sentence in the manuscript may mislead the readers regarding the relevance of homogeneous guide fields in nature. Consequently, we clarified it by modifying the paragraph as follows:

“There is however no clear reason why the out-of-plane magnetic field would be uniform in **some** natural events and not more alike our experimental setup (see e.g. Fig.5 in Ref.). **Although uniform guide-fields are relevant to some reconnection events [Eriksson et al., 2016]**, previous numerical studies used a uniform guide field as a numerical simplification more than a necessary property.”

While the revised text is much improved, there is still room for additional clarification. For example, the second paragraph of the Introduction begins as follows: “A factor complicating the picture is the topology of the magnetic fields. Deviating from the idealized picture of the first theoretical models [Refs], in which investigations were restricted to two-dimensional (e.g. in the yz plane) anti-parallel magnetic fields, the magnetic fields in natural three-dimensional events [Refs] can be decomposed into anti-parallel and possibly an additional component (e.g. along the x–axis) of the magnetic field (commonly called a guide field).” As written, this description could be interpreted as implying that a 3D geometry is necessary for the inclusion of a guide field. I’m sure the authors did not mean to suggest this to be the case since the orientation of B in a 2D simulation is not restricted to lie in the simulation plane. Please reword these sentences to avoid any ambiguity on this point.

Thank you for the suggestion. In order to remove possible ambiguity here, we have rephrased the sentence as follows:

“A factor complicating the picture is the topology of the magnetic fields. Deviating from the idealized picture of the first theoretical models\cite{Parker1957}, in which investigations were restricted to the **reconnection of anti-parallel magnetic fields (e.g. the components of the incoming magnetic field lie** in the yz plane), the magnetic fields in natural events\cite{Trattner2007a,Qiu2010} can be decomposed into

anti-parallel components and possibly an additional component (e.g. along the x -axis) of the magnetic field (commonly called a guide field).”

The same paragraph ends as follows: “The pending challenge the community faces here is to understand how does reconnection take place between two obliquely oriented flux tubes, i.e. in a guide-field reconnection configuration.” This sentence suggest that the “guide-field reconnection configuration” is the defining property of obliquely oriented flux tubes rather than one of many characteristic properties. This distinction could be made more clearly by replacing “i.e.” with a more flexible transition (for example, “. . . flux tubes, which includes the presence of a guide field.”).

We thank the reviewer for the suggestion and we concur with this suggestion. The sentence has been modified as suggested.

In summary, this manuscript now clearly presents the key experimental results and their relevance to a particular domain of naturally occurring plasma interactions without overstating the generality of the finding to all scenarios in which guide-field reconnection can occur. As pointed out, there remain a few cases where minor revisions to the wording would help to further avoid potential misunderstanding or ambiguity. As previously mentioned, I also believe that a few additional figures (3D volume rendered views of the simulation results similar to Fig. 1d as well as additional panels in Fig. 6 for a larger value or values of x) would aid the reader in visualizing the inhomogeneous nature of the interacting plasma plumes. Pending these modifications, I believe this work should meet the criteria for publication in Nature Communications.

We thank the reviewer for his/her encouraging comments. We hope that the paper is more clear with these last modifications.

REVIEWERS' COMMENTS

Reviewer #1 (Remarks to the Author):

The latest revisions to the manuscript adequately address the concerns raised in my previous report.

The new panels of Fig. 6 for $x=25 d_0$ are particularly helpful. Specifically it is quite interesting to see that the current sheet appears even more coherent for $\theta = 0^\circ$ at $x=25 d_0$ (panel g) than at $x=10 d_0$ (panel d), which is the opposite of the case for the two nonzero tilt angles. Please note, however, that the caption for this figure as it refers to these new panels needs to be corrected -- as both the value of x ($20 d_0$ instead of $25 d_0$) and the panel references (d-f instead of g-i) are in error.

Also, in the new paragraph on p.2 beginning "Note that the effective distance...", please indicate that the 7% increase is specifically for $x=25 d_0$.

I am now satisfied with the manuscript as revised (with these very minor revisions), and can now recommend its publication in "Nature Communications."

Answers to the comments made by Reviewer 1

Reviewer #1 (Remarks to the Author):

The new panels of Fig. 6 for $x=25 d_0$ are particularly helpful. Specifically it is quite interesting to see that the current sheet appears even more coherent for $\theta = 0^\circ$ at $x=25 d_0$ (panel g) than at $x=10 d_0$ (panel d), which is the opposite of the case for the two nonzero tilt angles. Please note, however, that the caption for this figure as it refers to these new panels needs to be corrected -- as both the value of x ($20 d_0$ instead of $25 d_0$) and the panel references (d-f instead of g-i) are in error.

Thanks for spotting this. It is now corrected.

Also, in the new paragraph on p.2 beginning "Note that the effective distance...", please indicate that the 7% increase is specifically for $x=25 d_0$.

This is now indicated explicitly.

I am now satisfied with the manuscript as revised (with these very minor revisions), and can now recommend its publication in "Nature Communications."